# BBDetector: Intelligent border binary detection in IoT device firmware based on a multidimensional feature model

**Shudan Yue**[1,2], **Guimin Zhang**[1,2*], **Qingbao Li**[1], **Wenbo Zhang**[1],
**Xiaonan Li**[1,3], **Weihua Jiao**[1]

**1** Information Engineering University, Zhengzhou, China, **2** Laboratory for Advanced Computing and Intelligence Engineering, Wuxi, China, **3** School of Computer Science, Zhongyuan University of Technology, Zhengzhou, China

\* zh.guimin@163.com

**Data availability statement:** The code supporting this study's findings has been made

## Abstract

In the field of firmware security analysis for Internet of Things (IoT) devices, border binary detection has become an important research focus. However, the existing methods for border binary detection have problems such as insufficient feature characterization, high false-negative rates, and low intelligence levels. To mitigate these issues, we introduce BBDetector, a border binary detection method based on a multidimensional feature model. First, we constructed the first known set of border binaries at a certain scale by collecting and analyzing a diverse set of real-world firmware. To characterize the features of border binaries comprehensively, we proposed a multidimensional feature model (MDFM). Next, we extracted the feature vectors of binaries via the MDFM and designed a novel stacking method to achieve border binary detection. This method involves ensemble learning, combining extreme gradient boosting, light gradient boosting machine, and categorical boosting as base learners with random forest as the meta-learner. Finally, a border binary detection model (XLC-R) was obtained by training with feature vectors. We tested and evaluated BBDetector on two datasets. The experimental results showed that XLC-R achieved a precision of 94.98%, a recall of 91.02%, and an F1 score of 92.84% for the constructed representative Dataset I. Additionally, BBDetector detected 3.25 times and 2.23 times more border binaries in Dataset II than did the state-of-the-art tools Karonte and SaTC, respectively. BBDetector provides an accurate method for border binary detection in IoT firmware security analysis, significantly enhancing the pertinence of vulnerability detection, dramatically reducing the complexity of firmware security analysis, and providing essential technical support for improving IoT device security.

## 1 Introduction

With the continuous advancement of Internet of Things (IoT) technology, IoT devices have become indispensable components in many fields, such as industrial automation,

publicly available without restrictions. It is hosted on GitHub (https://github.com/YueShudan/BBDetector) and shared under an open-source License to facilitate reproducibility and reuse.

**Funding:** This research was supported by the fund of Laboratory for Advanced Computing and Intelligence Engineering (2023-LYJJ-01-032) and the National Key Research and Development Program of China (2021YFB3101804). The funders had no role in study design, data collection and analysis, decision to publish, or preparation of the manuscript. There was no additional external funding received for this study.

**Competing interests:** The authors have declared that no competing interests exist.

transportation systems, smart homes, and smart cities. The number of IoT devices worldwide is projected to reach 32.1 billion by 2030 [1]. These devices interact and communicate frequently with the outside world over networks, which, although offering conveniences in daily life, raises concerns regarding security risks. Currently, the number of network attacks on IoT devices in the real world is increasing [2,3].

Linux-based firmware images are crucial for IoT devices because of their widespread deployment [4–6]. However, if the components of these firmware are exploited by attackers, there may be severe consequences. Most IoT firmware contains many complex and diverse interrelated components, making comprehensive security analysis a challenging task [7,8]. To reduce the complexity of such an analysis, most studies [9,10] are based on the subjective experiences of analysts when selecting the binary objects to be analyzed. Unfortunately, owing to the limitations of researchers' experiences, these methods have limitations in identifying other important binary executables.

Notably, a class of firmware binaries is responsible for receiving and processing external input data according to predefined rules and logic. These binaries act as interfaces between devices and their external environments. The security and stability of these binaries are crucial for the overall operation of the entire device. Researchers first introduced the concept of 'border binaries' to describe this type of binary and developed Karonte to address the detection of such border binaries [11]. Through cross-binary taint analysis, Karonte identified a total of 46 zero-day bugs, highlighting the importance of border binaries for firmware security. In subsequent studies, SaTC [12], Sinktaint [13] and HermeScan [14] were developed, which fully utilize border binaries to achieve better analysis results. On the basis of these studies, we focus on border binary detection in IoT device firmware in this paper.

To detect border binaries more accurately, Karonte [11] examines the features of network parsing functions and uses a density clustering algorithm to identify border binaries. SaTC [12] locates border binaries by identifying common string literals that are shared between front-end files and back-end binaries. SinkTaint [13] utilizes the front-end and back-end shared strings to identify border binaries and finds more candidate functions that receive user input through the parameter parser and implicit keyword identification. HermeScan [14] locates border binaries by relying on SaTC and leverages fuzzy matching to identify candidate functions receiving user input. Although these heuristics automatically implement border binary detection, the number of false negatives is still high.

Our intuition is that binaries with similar functionality generally have similar semantic and behavioral features, such as loop behaviors and application programming interface (API) calls. After analyzing numerous binaries, we discovered that although border binaries are complex and diverse, they can still be generalized and represented on the basis of a limited number of features. Hence, this paper aims to develop an intelligent border binary detection method that does not rely on the source code and further reduces false negatives.

Currently, there are three challenges in detecting border binaries intelligently.

**C1: Selection of border binary features.** Different IoT devices may have different border binaries, which vary depending on the network protocol used. Existing solutions, such as Karonte [11] and SaTC [12], exhibit notable limitations in feature selection: Karonte [11] utilizes only five features for detecting border binaries, while SaTC [12] relies on shared keywords between the web front-end and back-end. These methods do not fully capture the essential features of the border binaries, leading to inaccurate detection results. To date, there is no clear standard to guide feature selection and extraction for border binaries. Therefore, determining which features can accurately and comprehensively characterize the border binaries is the primary challenge in achieving accurate detection.

**C2: Construction of a border binary dataset.** Constructing a high-quality border binary dataset is crucial for improving the robustness of border binary detection models. Existing solutions [14–16] either rely on heuristic methods to identify border binaries without using datasets, or rely on small-scale datasets, failing to allocate sufficient resources to build widely used border binary sets. Given the current lack of open-source border binary datasets for training machine learning (ML) models, it is necessary to construct such a dataset through automated tools and manual review. However, this part of the work heavily relies on expert experience and requires significant human resources.

**C3: Choice of a border binary detection model.** Existing solutions [11,17] perform poorly in identifying diverse border binaries. For example, rule-based methods [12,17] struggle to adapt to the protocols of diversified devices, while traditional machine learning-based solutions [11,14,16] exhibit limited generalization ability when encountering unseen border binaries. Therefore, selecting an appropriate border binary detection model is crucial for achieving higher accuracy and improving the efficiency of border binary detection.

To address the above challenges, we present BBDetector, a border binary detection method based on a multidimensional feature model. First, we collected numerous border binaries through a man-machine combination analysis method to construct the first known border binary dataset. After many samples were analyzed, we constructed a multidimensional feature model (MDFM), which includes program-level, function-level structural, behavioral, and semantic features for characterizing border binaries. Then, border and non-border binary features were extracted via the MDFM to construct feature vectors, which were preprocessed by PowerTransformer [18] to enhance the feature value distribution and improve model stability. Finally, we used extreme gradient boosting (XGBoost) [19], light gradient boosting machine (LightGBM) [20], and categorical boosting (CatBoost) [21] as the base classifiers and random forest (RF) [22] as the metaclassifier to train a stacking model (XLC-R) for border binary detection. XLC-R has the ability to detect border binaries in unknown firmware.

We implemented BBDetector and evaluated it on two datasets. The detection model (XLC-R) of BBDetector was trained and validated with a set of 320 representative firmware samples from 67 unique device series, which were selected from an exhaustive review of 8,192 firmware samples from 10 vendors. The results show that XLC-R achieves a precision of 94.98% and a recall of 91.02%. Additionally, we used 49 Linux-based firmware samples from the Karonte dataset as a benchmark to compare BBDetector with Karonte and SaTC. We demonstrated that BBDetector successfully identified 3.25 times and 2.23 times more border binaries than did Karonte and SaTC, respectively.

The main contributions of the paper are summarized as follows:

- We construct the first dataset for border binary detection. The dataset includes 11 network protocols encompassing a wide range of device types. This border binary dataset serves as a crucial foundation for further research in this field.
- We propose a multidimensional feature model (MDFM), which accurately characterizes the features of border binaries and provides an important reference for border binary detection research.
- We design a border binary detection model (XLC-R), which uses a stacking method of ensemble learning to achieve efficient border binary detection.
- We implement an intelligent border binary detection method (BBDetector) and evaluate it with different datasets. The results showed that BBDetector achieved better performance in border binary detection.

The remainder of this paper is organized as follows. Sect 2 summarizes related work for BBDetector. Sect 3 introduces the motivation for this study. Sect 4 outlines the framework of BBDetector, and its design is described in Sect 5. Sect 6 presents our implementation and the results of evaluating BBDetector on real-world firmware samples. Sect 7 discusses the limitations of this study and future research directions. Sect 8 concludes this paper and outlines the future work.

## 2 Related work

An analysis of existing research shows that methods for identifying critical binaries in IoT firmware can be divided into three categories. The first category consists of approaches that rely on expert knowledge, where researchers manually select potential critical binaries (such as *httpd* and *cgibin*) as objects for security analysis [5,9,10,23–25]. The second category consists of approaches that are based on heuristic methods, automatically identifying critical firmware binaries as objects through feature analysis [11–15]. The third category consists of approaches that dynamically emulate the external communication services [26–29] received in IoT firmware. However, the first category of approaches relies too heavily on expert knowledge, while the third category of approaches requires device emulation, which demands significant computational resources and poses challenges in firmware emulation. This paper primarily focuses on the second category of method, which involves automated identification.

Karonte [11] uses a variety of features, including the numbers of basic blocks, branches, network-encoding strings, conditional statements used in combination with memory comparisons, and whether any data read from network sockets is used in a memory comparison, to calculate the scores of network parsing functions in binaries. The density-based clustering with noise (DBSCAN) algorithm is subsequently used to identify the initial border binary set with higher scores of network parsing functions. This method determines border binaries by identifying the network parsing function and targets mainly the border binary related to the HTTP protocol. SaTC [12] identifies border binaries by determining critical functions on the basis of shared strings between the front file and back binaries. The underlying principle is that a border binary often receives and processes the data transmitted by the front-end using strings from the front-end file as the label. Hence, identifying the shared string can preliminarily determine the border binary. The approach is efficient for web services implemented by the MQTT and UPnP protocols. FBI [15] uses vulnerability-related binaries and vulnerability information collected from the internet as clues for analyzing insecure firmware images. This approach leverages existing vulnerability knowledge, primarily identifying binaries that may have similar vulnerabilities. However, its ability to detect border binaries in firmware is limited. SinkTaint [13] adopts SaTC that identifies border binaries using shared strings between the front-end and back-end, and further finds more candidate functions that receive user input by the parameter parser and implicit keyword identification. HermeScan [14] further optimizes SaTC [12] and performs fuzzy matching of the shared strings between the front end and the discovered border binaries to identify their critical functions. This method once again demonstrates the importance of the string information for border binary detection.

Network parsing functions are an important component of border binaries. Thus, some methods for identifying network parsing functions can also be used for border binary detection. PIE [17] summarizes a set of features specific to parsing functions in embedded systems and uses a heuristic algorithm and a cross-validation technique to identify unknown firmware parsing functions. Zheng et al. [16] used the features of protocol parsing functions to identify these functions through a support vector machine (SVM) classifier. The approach of identifying protocol parsing functions can be used to locate border binaries in firmware. This

method emphasizes the importance of the protocol parsing functions. However, the small sample set used for training and validation results in low accuracy in recognizing the protocol parsing functions. Furthermore, HumIDIFy [30] leverages strings and the function import and export tables information as features and uses a semi-supervised machine learning (ML) method to identify the functionalities of binaries such as web servers, FTP daemons, and Telnet daemons. Although this approach serves as a valuable reference for program-level feature extraction, its reliance solely on these features may yield numerous false positives.

Furthermore, the characterization of object features [31–34] and the construction of detection models [35–43] are also essential factors influencing border binary detection. For example, Gemini [31] transforms control flow graphs into numeric feature vectors and uses a graph neural network (GNN) to implement binary vulnerability detection. Li et al. [35] proposed an intrusion behavior detection method based on an ensemble approach, which introduces a leader class and confidence decision ensemble. Saheed et al. [33] combined an autoencoder with modified particle swarm optimization for feature selection and utilized a deep neural network (DNN) for attack detection. Qiao et al. [41] used a multihead attention self-supervised representation model to achieve industrial sensor anomaly detection. Li et al. [32] integrated a composite feature model with graph neural networks to achieve function similarity detection. Saheed et al. [42,43] employed optimized genetic algorithms and long short-term memory networks for detecting relevant targets. Hou et al. [34] proposed a network intrusion detection method based on hierarchical dependencies and class imbalance. These methods leverage different features and various machine learning models to attain accurate target recognition, providing significant insights for the method proposed in this paper. However, due to the diversity of IoT protocols and the complexity of firmware behavior, existing methods cannot be directly applied to border binaries detection in IoT devices. Therefore, further targeted improvements and optimizations are necessary, considering the unique characteristics of border binaries.

In summary, existing border binary detection methods have significant room for improvement regarding feature characterization, false negative rates, and intelligence levels. To address the issue of insufficient feature characterization, we designed a multidimensional feature model that incorporates both program-level and function-level features. These features capture the behavior of border binaries from multiple dimensions, significantly enhancing the accuracy and comprehensiveness of feature characterization. To tackle the problem of high false negative rates, we constructed a border binary dataset containing multiple common protocols. By analyzing a large number of border binaries, the dataset's diversity and representativeness are ensured, thereby improving the detection capability of the model in different protocol scenarios. To enhance the intelligence level, we adopted a stacking method based on ensemble learning, which combines XGBoost, LightGBM, and CatBoost as base learners, with RF as the meta-learner. By stacking multiple machine learning models for border binary prediction, BBDetector can more effectively learn the key feature attributes of border binaries, further enhancing detection accuracy and robustness.

## 3 Motivation

The security vulnerabilities of IoT devices are predominantly due to their unsecured communication with cloud platforms and mobile devices. Compared to offline devices, IoT devices have more complex attack surfaces [44,45]. Given the increasing complexity of IoT device firmware, researchers usually adopt an analysis strategy based on expert knowledge to improve the efficiency of firmware security analysis, prioritizing or only selecting common web servers (such as *httpd*, *prog.cgi*, and *netcgi*) as the main objects to be analyzed. This type

of method is limited by researchers' experience and ability, and easily overlooks the analysis of certain important binaries. The border binary concept was proposed to guide the security analysis process.

Border binaries are the executable binaries within IoT firmware that directly interact with external network data. They are responsible for receiving, parsing, and processing network requests from external sources, then passing the results to other modules within the firmware for further processing. As critical components of firmware, the importance of border binaries is reflected in two key aspects. First, they are involved not only in fundamental network communication operations, such as receiving and sending data, but also in handling critical tasks like identity authentication and firmware updates, playing a vital role in the functionality of the firmware. Second, due to the widespread vulnerabilities inherent in border binaries, they have become the primary entry point for most malicious data aiming to infiltrate the firmware, posing a significant threat to firmware security. Therefore, accurately identifying border binaries is crucial for ensuring the security of IoT devices, as the vulnerabilities in these binaries could allow attackers to breach the device's defenses, infiltrate the firmware, and severely compromise the security of the entire IoT system.

IoT devices receive and process external requests via various typical network protocols [46,47]. As an example, the Universal Plug and Play (UPnP) protocol is designed to enable devices in a home network to easily discover and communicate with each other without requiring complex user configurations. Fig 1 shows the decompiled code of the function *genacgi_main* in the binary *cgibin* of *D-Link DIR-822A1_FW103WWb03*. The function *genacgi_main* processes HTTP requests from clients via environmental variables such as REQUEST_METHOD, REQUEST_URI, and SERVER_ID, and parses the request parameters based on the request types for achieving SUBSCRIBE and UNSUBSCRIBE requests related to the UPnP protocol. Notably, the script path *htdocs/upnp/run.NOTIFY.php* is hardcoded directly in the function *genacgi_main*. The *run.NOTIFY.php* script calls the function *GENA_subscribe_new* in a PHP script and passes the result to the variable SHELL_FILE, which is obtained by the function *genacgi_main*, resulting in the unauthenticated remote code execution vulnerability CVE-2019-17621 [48].

In the above example, *cgibin* is a border binary, and *genacgi_main* is a network parsing function tasked with receiving and processing external network inputs. If an attacker attempts to enter the firmware, they must go through the network parsing function in the border binary. Deficiencies in border binary design can easily lead to potential security issues.

As the first line of defense for firmware when interacting with external network data, the border binary is not only responsible for the initial processing of communications but also involves critical security processes such as identity authentication and firmware upgrades. However, because different device vendors may use different communication protocols and interface designs when developing device firmware, identifying border binary becomes very challenging.

At present, although research on border binary detection methods [11,12] has made some progress, these approaches still have certain limitations. One key issue is insufficient feature characterization. Karonte uses only five features to identify border binaries, and SaTC relies solely on shared strings between the front and back ends of web services. These approaches may not fully capture the important features of border binaries during the feature selection process. Another issue is the high false-negative rates. Karonte targets mainly the HTTP protocol, and SaTC cannot detect cases where data are not transmitted through shared strings, such as data achieved through protocol interaction.

```
int genacgi_main(void) {
    char *request_method = getenv("REQUEST_METHOD");
    char *request_uri = getenv("REQUEST_URI");
    ...// retrieves the HTTP request in the environment variable
    ...//parsing request
    request_uri = strstr(request_uri, "?service=") + 9;
    if (strcmp(request_method, "SUBSCRIBE") == 0 && http_sid &&
    ↪  http_callback) {
        http_callback[strcspn(http_callback, ">")] = '\0';
        if (*http_callback == '<') http_callback++;
        if (strncmp(http_callback, "http://", 7) == 0) {
            char *host = strchr(http_callback + 7, '/');
            if (host) {...
                snprintf(command, sizeof(command), ...,
                ↪  "/htdocs/upnp/run.NOTIFY.php\n METHOD=SUBSCRIBE\n
                ↪  INF_UID=%s \nSERVICE=%s\n HOST=%s\n URI=/%s\n
                ↪  TIMEOUT=%d\n REMOTE=%s\n
                ↪  SHELL_FILE=/var/run/%s_%d.sh", ...);
            xmldbc_ephp(0, 0, command, stdout); ...}
        }
    } else if (strcmp(request_method, "UNSUBSCRIBE") == 0 && http_sid)
    ↪  {
    // Handle the UNSUBSCRIBE logic
    } else {cgibin_print_http_status(400, "Invalid Request");}
    return 0;
}
```

**Fig 1. Decompiled code of a network parsing function of a border binary (simplified).**

Therefore, we aim to develop a novel automatic detection method to comprehensively and accurately identify border binaries in IoT firmware and provide important technical support and methodological guidance for enhancing security analysis of IoT firmware.

## 4 Overview

The general framework of BBDetector is shown in Fig 2. BBDetector consists of four modules: firmware unpacking, border binary collection, feature extraction, and detection model training and prediction.

**Firmware unpacking.** BBDetector unpacks firmware samples via the unpacking utility *binwalk* [49] and automatically retrieves file attributes such as file type, permissions, and extensions. Given that border binaries exist only in binary executables, BBDetector builds a series of filtering rules on the basis of these attributes to systematically exclude scripts, configuration files, dynamic link libraries, etc.

**Border binary collection.** Many candidate border binaries and network parsing functions are obtained through the optimized Karonte [11]. Reverse engineering tools are used to manually confirm the border binaries (refer to Sect 5.2 for more details).

**Feature extraction.** The feature vectors of binary executables are extracted via the MDFM. Specific details on the MDFM are provided in Sect 5.3.

**Detection model training and prediction.** First, BBDetector uses PowerTransformer to preprocess the feature vectors (Sect 5.4). A stacking model (XLC-R) is subsequently constructed for better classification (Sect 5.5). Finally, XLC-R implements intelligent border binary detection.

The workflow of BBDetector is divided into two phases: the training phase and the detection phase. In the training phase, we first use the collected border and non-border binaries

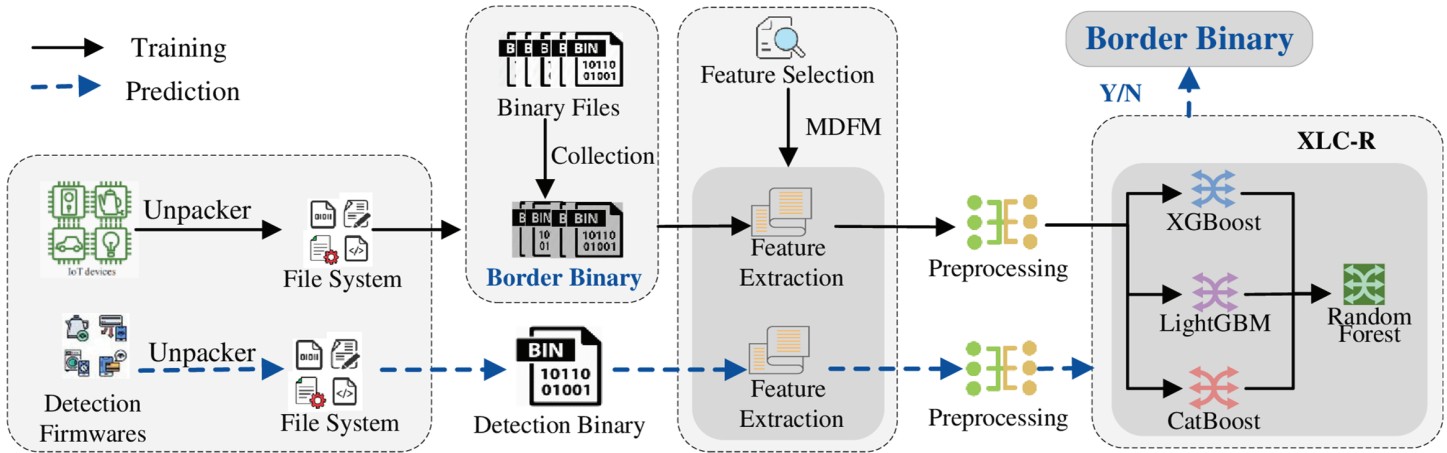

**Fig 2. Framework of BBDetector.**

to construct a sample set for training and validation (Dataset I in Sect 6.3). Then, the feature vectors of all samples are extracted based on MDFM, and the PowerTransformer is used for data preprocessing. Finally, XLC-R is trained via the stacking method. In the detection phase, an entire firmware image is input into BBDetector. First, BBDetector unpacks the firmware and extracts the binary executables in the firmware. Then, all feature vectors of the binary executables are extracted based on MDFM and normalized. Finally, the normalized feature vectors are input into XLC-R for prediction, and the feature vectors belonging to the border binaries are obtained, which correspond to the border binaries.

## 5 Design

By leveraging the framework described in Sect 4, we design and develop BBDetector to enable efficient border binary detection.

### 5.1 Firmware collection and unpacking

Typically, IoT firmware can be downloaded from vendor websites, captured during OTA updates, extracted from hardware, or obtained from third-party archives [27]. We successfully collected 12,893 firmware samples from 11 vendors through web crawling.

To conserve storage space, vendors compress firmware images in various formats (e.g., .BIN, .ZIP). We use *binwalk* [49] to unpack firmware images and extract their file systems. However, there is an issue of incomplete extraction when *binwalk* extracts the JFFS2 and SquashFS file systems. Referring to the solutions proposed in FirmSec [44], we use Jefferson and Sasquatch to replace jffsdump and unsquashfs, respectively, in *binwalk*, thereby achieving complete extraction of the JFFS2 and SquashFS file systems.

We successfully extracted 8,192 file systems from the 12,893 IoT firmware collected.

### 5.2 Border binary collection

When IoT firmware interacts with the outside world, it relies on various network protocols. Therefore, we analyzed the typical protocols used by IoT firmware, including UPnP, RTMP, SMTP, PPP, SOAP, HNAP, SSDP, CWMP, HTTP, MQTT, and DAAP. By analyzing known

network parsing functions massively in border binaries, we found that border binaries usually have the following characteristics:

- Border binaries usually call specific APIs to receive and parse user input data, such as the examples of RECV_API in Table 1;
- Border binaries usually use specific strings that are correlated with the protocol type, such as the examples of COMM_STR in Table 1;
- Border binaries use many APIs related to string comparison, string search, and string conversion to call the corresponding APIs, such as the examples of STR_API in Table 1.

Based on the analysis of border binary features, we integrated the network-encoding strings (refer to the COMM_STR examples in Table 1) related to these protocols into Karonte's list of network-encoding strings. Additionally, border binaries often call special APIs to process external input data. Therefore, we collected a list of such APIs (refer to the RECV_API examples in Table 1). Given that Karonte does not use the RECV_API feature, we also optimized Karonte's feature extraction process to successfully introduce the RECV_API examples. Finally, we modified the calculation score $ps_j$ of the network parsing function in Karonte [11], and the new score $newps_j$ calculation is shown as:

$$newps_j = ps_j * (1 + 0.2 * \#RECV\_API) \tag{1}$$

Through the optimized Karonte, we obtained many candidate network parsing functions from 8,192 firmware samples and then determined whether the binary executables in which they are located are border binaries on the basis of the candidate network parsing functions.

To maximize the accuracy of the samples, we rigorously screened and confirmed each border binary by reverse engineering. Our analysis revealed three primary issues:

- **Similarity across firmware versions and series.** The binary executables in most firmware samples of the same models and different versions were roughly the same (e.g., *D-Link*

**Table 1. List of features and their examples.**

| Feature | | Example |
|---------|---------|---------|
| RECV_API | NET_API | recv, recvfrom, connect, listen, bind, accept, socket, send, sendto, sethostbyname, getaddrinfo, getnameinfo |
| | I/O_API | gets, getchar, sscanf, scanf, fgets, fget, fgetc, fscanf, getline, open, fopen, fread, fclose, read, readfile |
| | ENV_API | getenv, getenv_s, setenv |
| COMM_STR | AUTH_STR | username, password, UserPwd, Authenticate-Request, Authenticate-Ack, Authenticate-Nak, Challenge, Response, Success, Auth, login |
| | HTTP_STR | GET, POST, PUT, DELETE, HEAD, CONNECT, REQUEST_METHOD, QUERY_STRING, boundary=, HTTP_, query, Content-Type, CONTENT_TYPE, ACCEPT, socket, user-agent, request |
| | MQTT_STR | MQTT, QoS, PINGREQ, PINGRESP, CONNACK, publish, topic, SUBSCRIBE, PUBLISH, SUBACK, PUBACK, client_topic |
| | CONN_STR | REMOTE_ADDR, RemoteEndPoint, Client_id, server_host, ProtocolVersion, admin., getserverHost, URL, soap, index., remote, SSID |
| STR_API | STRCMP_API | strcmp, strncmp, strlcmp, strcasecmp, strncasecmp, memcmp, stricmp |
| | STROP_API | strlen, strdup, strsep, strtok, strcat |
| | STRSEARCH_API | strchr, strstr, strrchr, strcsnp |
| | STRCONVER_API | strtol, strtoll, atoll, atol, atoi |

*DCS930l_v109* and *D-Link DCS930l_v111*), and even some firmware samples of the same series and different models were also largely similar (e.g., *D-Link DIR-890L* and *D-Link DIR-895L*).

- **False positives.** According to the verification results, some candidate border binaries were false positives that did not meet the definition of border binaries.
- **False negatives.** Through empirical analysis, we identified several other border binaries in addition to candidates. There is no obvious network parsing functionality in these border binaries, which are implemented by multiple functions (For example, the network parsing functionality of the binary upnp in *Linksys E800-v1.01.004* firmware consists of several functions such as *parse_uri*, *parse_csghdr*, *soap_comtrol*, *soaps_process*, etc).

Finally, by removing duplicate and false-positive candidate functions and adding various manually confirmed network parsing functions, a total of 1,307 network parsing functions were selected from 320 representative firmware samples from 67 different device series from 10 vendors over three months. These network parsing functions correspond to 1,103 different border binaries. Notably, some border binaries contain more than one network parsing function.

## 5.3 MDFM

In the process of constructing the border binary dataset, we have summarized two viewpoints regarding border binary features: Firstly, network parsing functions are crucial features of border binaries, and binaries that possess network parsing functions can be essentially identified as border binaries. Secondly, a few border binaries lack specific network parsing functions, as some binaries do not have relatively independent network parsing functions and instead depend on a combination of multiple functions to fulfill network parsing, which is less common.

Based on the aforementioned insights, we believe that an effective feature model for border binary should have the following descriptive abilities:

- The ability to characterize network parsing functions, in order to better characterize border binary features containing explicit network parsing functions.
- The ability to characterize the overall binary features, while also considering the border binary features without obvious network parsing functions.
- The feature weight of network parsing functions should be heavier than the features of the overall binary, in order to reflect the situation where the probability of border binary without explicit network parsing functions is relatively small.

On the basis of the above three guiding strategies, we constructed a multidimensional feature model (MDFM) for comprehensively characterizing border binaries. MDFM = $[V_0, V_1, V_2, ..., V_{14}]$. The MDFM is composed of 15-dimensional vectors, which include program-level behavioral features ($V_0$) and semantic features ($V_1$), as well as function-level structural features ($V_2$, $V_3$), behavioral features ($V_4 \sim V_{10}$), and semantic features ($V_{11} \sim V_{14}$).

The feature selection of MDFM is grounded in two perspectives. First, the network parsing function serves as a key feature of the border binaries, as it directly reflects the interaction behavior between the binaries and the external network. Second, while some border binaries may lack explicit network parsing functions, they often depend on multiple functions to collectively achieve network parsing functionality. As a result, MDFM characterizes border binaries from multiple dimensions through program-level and function-level features, ensuring the model comprehensively covers all possible border binaries.

The 15-dimensional features characterize border binaries from different perspectives and exhibit the following complementary relationships:

- Program-level features ($V_0$ and $V_1$) provide a global perspective, aiding in identifying the binary's overall behavior and semantics. For example, $V_0$ can preliminarily assess whether the binary may be a border binary by counting the number of APIs related to external input in the function import table.
- Function-level features ($V_2 \sim V_{14}$) provide a localized perspective, aiding in identifying specific network parsing functions.
- Feature weight allocation. In the feature of border binaries, function-level features are assigned higher weights, reflecting the importance of network parsing functions in border binaries, while program-level features act as supplementary components to ensure that the model encompasses all possible border binaries.

By utilizing the complementary design of multidimensional features, MDFM accurately and comprehensively characterizes border binary features, enhancing the accuracy and robustness of border binary detection. A detailed list of the feature vectors and their corresponding descriptions is provided in Table 2.

$V_0$: **Number of APIs in binaries related to receiving external input** (RECV_API). Since APIs related to receiving external input serve as direct credentials for border binaries to interact with external networks, we analyzed the function import table of border binaries and discovered that they typically call specific APIs when receiving and processing external input data. Therefore, we present examples of RECV_API, as shown in Table 1.

$V_1$: **Number of strings in binaries related to network communication** (COMM_STR). The communication of IoT devices involves four main stages: pairing, binding, authentication, and control [50]. The communication-related strings (such as HTTP, MQTT, etc.) are important indicators of the border binaries. When border binaries interact with the outside world, they usually need to match various hardcoded strings related to communication to process requests accurately. Therefore, we provide examples of COMM_STR, as illustrated in Table 1.

**Table 2. Feature vectors of the MDFM and their corresponding descriptions.**

| Feature vector | Description |
|---|---|
| $V_0$ | Number of APIs related to receiving external input (RECV_API) |
| $V_1$ | Number of strings related to network communication (COMM_STR) |
| $V_2$ | Number of Basic Blocks |
| $V_3$ | Number of Branches |
| $V_4$ | Number of APIs related to network communication (NET_API) |
| $V_5$ | Number of APIs related to Standard I/O (I/O_API) |
| $V_6$ | Number of APIs related to Environment variable (ENV_API) |
| $V_7$ | Number of APIs related to string comparison (STRCMP_API) |
| $V_8$ | Number of APIs related to string operation (STROP_API) |
| $V_9$ | Number of APIs related to string search (STRSEARCH_API) |
| $V_{10}$ | Number of APIs related to string conversion (STRCONVER_API) |
| $V_{11}$ | Number of strings related to authentication (AUTH_STR) |
| $V_{12}$ | Number of strings related to HTTP requests (HTTP_STR) |
| $V_{13}$ | Number of strings related to MQTT requests (MQTT_STR) |
| $V_{14}$ | Number of strings related to C/S connection requests (CONN_STR) |

$V_2$: **Number of basic blocks in the function**. Due to the fact that network parsing functions are responsible for handling data received from external sources. Their internal workings are usually complex, comprising many control flows and jump operations. Therefore, if the number of basic blocks in a function is within a certain threshold range, then the function is likely to be a network parsing function of the border binary to implement processing functionality.

$V_3$: **Number of branches in the function**. Conditional branch statements are commonly used to address branching logic and loop logic, which represent the control flow complexity of a function. Since network parsing functions are typically implemented as the protocol's state machines and must adhere to the communication rules of network services to influence control flow decisions, they are expected to contain a certain number of conditional branch statements. Therefore, we consider the number of conditional branch statements in a function as a structural feature of border binaries.

$V_4 \sim V_6$: **Number of APIs in the function related to receiving external input** (NET_API, I/O_API, and ENV_API). Considering that network parsing functions call APIs related to processing inputs from external sources when handling external data. Therefore, we consider the number of APIs that receive external input in the function as behavioral features of the border binary. Furthermore, inspired by SDCFM [32], we classified RECV_API to better characterize the functionality of these functions. The number of network communication-related APIs (NET_API) is denoted as $V_4$. The number of standard I/O-related APIs (I/O_API) is denoted as $V_5$. The number of environment-related APIs (ENV_API) is denoted as $V_6$. Examples of NET_API, I/O_API, and ENV_API are given in Table 1.

$V_7 \sim V_{10}$: **Number of APIs in the function related to manipulating strings** (STRCMP_API, STROP_API, STRSEARCH_API, and STRCONVER_API). When border binaries interact with the outside world, network parsing functions need to validate the input data (for example, check the length, type, and range), which involves data format conversion, string splitting, merging, matching, and replacing operations. Therefore, we consider the number of APIs related to string operations in the function as a behavioral feature of the border binary. The number of string comparison-related APIs (STRCMP_API) is denoted as $V_7$. The number of string operation-related APIs (STROP_API) is denoted as $V_8$. The number of string search-related APIs (STRSEARCH_API) is denoted as $V_9$. The number of string conversion-related APIs (STRCONVER_API) is denoted as $V_{10}$. Examples of STRCMP_API, STROP_API, STRSEARCH_API, and STRCONVER_API are given in Table 1.

$V_{11} \sim V_{14}$: **Number of strings in the function related to network communication** (AUTH_STR, HTTP_STR, MQTT_STR, and CONN_STR). In border binaries, network parsing functions compare the received data with a preset list of network-encoding strings stored in memory, according to the requirements of the relevant protocol. Therefore, we take the number of communication-related strings in the function as a semantic feature of the border binary. The number of authentication-related strings (AUTH_STR) is denoted as $V_{11}$. The number of HTTP request-related strings (HTTP_STR) is denoted as $V_{12}$. The number of MQTT request-related strings (MQTT_STR) is denoted as $V_{13}$. The number of Client/Server connection-related strings (CONN_STR) is denoted as $V_{14}$. Examples of AUTH_STR, HTTP_STR, MQTT_STR, and CONN_STR are given in Table 1.

We utilized the reverse engineering tool *Radare2* [51] to extract corresponding features from the control flow and data flow of the binary. The code for feature extraction is shown in Algorithm 1. Suppose that binary set $B$ contains the number of $b$ is $n$, the average number of functions in each binary $b$ is $m$, and $k$ is the average number of feature extraction operations in each function $f$. The time complexity of Algorithm 1 mainly consists of the following parts: (1) outer loop: traversing each binary $b$ in binary set $B$, with time complexity of $O(n)$;

(2) Inner loop: Traversing each function $f$ in each binary $b$, with time complexity of $O(m)$; (3) Internal operation: performing feature extraction on each function $f$, including counting the number of basic blocks, branch statements, API calls, and string matches. Assuming the average number of these operations in each function $f$ is a constant $k$, the time complexity of the internal operations is $O(k)$. Therefore, the total time complexity of Algorithm 1 is $O(n{\times}m{\times}k)$.

**Algorithm 1. Feature extraction.**

```
Input: Binary Set B
Output: Feature Vectors of MDFM: V₀, V₁, ..., V₁₄
1  0 ← V₀, V₁, ..., V₁₄;  # Initialization Variable
2  for each binary b in B do
3      # Check the global import symbol table
4      V₀ ← COUNT(b.import_symbol_table ∩ RECV_API);
5      # Check the global string information
6      V₁ ← COUNT(b.string_information ∩ COMM_STR);
7      for each function f in b do
8          # Count the number of basic blocks and branch statements
9          V₂, V₃ ← f.cfg.block_count(), f.cfg.branch_count();
10         # Count the number of APIs related to receiving external input
11         RECV_API = [NET_API, I/O_API, ENV_API];
12         for i, recv_api in enumerate(RECV_API, start = 4) do
13             V[i] ← COUNT(f.lib_funcname ∩ recv_api);
14         end
15         # Count the number of APIs related to manipulating strings
16         STR_API = [STRCMP_API, STROP_API, STRSEARCH_API, STRCONVER_AP];
17         for i, str_api in enumerate(STR_API, start = 7) do
18             V[i] ← COUNT(f.lib_funcname ∩ str_api);
19         end
20         # Count the number of strings related to network communication
21         COMM_STR = [AUTH_STR, HTTP_STR, MQTT_STR, CONN_STR];
22         for i, comm_str in enumerate(COMM_STR, start = 11) do
23             V[i] ← COUNT(f.str_consts ∩ comm_str);
24         end
25     end
26 end
```

**An example.** We use the binary *httpd* and its network parsing function from the firmware *RT-AC51U* of the *ASUS* vendor to demonstrate a sample: [asus_RT-AC51U/httpd, main, 18476, 16, 46, 263, 178, 24, 11, 0, 1, 12, 8, 0, 61, 35, 36, 3]. Note that 174088 is the address of the *main* function. The following values are $V_0$, $V_1$, ..., and $V_{14}$, which represent the feature vectors for the MDFM of the border binary *httpd*.

## 5.4 Preprocessing

To prevent any single feature from disproportionately influencing the detection model due to variations in magnitude, proper preprocessing of feature vectors is important [52]. We first check the data type of the feature vectors, convert them into numerical values, and check for any missing values. For missing values, we fill them with 0. Then, we attempted five common preprocessing techniques: StandardScaler, MinMaxScaler, MaxAbsScaler, RobustScaler, and PowerTransformer (abbreviated as PowerTrans).

Table 3 shows the results of applying these five preprocessing methods to the binary *httpd* and its function *main* in the *ASUS RT-N15U* firmware, where PowerTransformer adopts the Yeo-Johnson transform. StandardScaler is sensitive to outliers and transforms the original feature *263* to *0.9*, causing extreme values to have a significant effect on the results. MinMaxScaler and MaxAbsScaler change both the original features *46* and *1* to *0.01*, losing the original range of the data. The reason why MinMaxScaler and MaxAbsScaler yield the same

**Table 3. Preprocessing results of a feature vector for *httpd* in the *ASUS RT_N15U* firmware.**

| Original Features | 16 | 46 | 263 | 178 | 24 | 11 | 0 | 1 | 12 | 8 | 0 | 61 | 35 | 36 | 3 |
|---|---|---|---|---|---|---|---|---|---|---|---|---|---|---|---|
| Preprocessing | $V_0$ | $V_1$ | $V_2$ | $V_3$ | $V_4$ | $V_5$ | $V_6$ | $V_7$ | $V_8$ | $V_9$ | $V_{10}$ | $V_{11}$ | $V_{12}$ | $V_{13}$ | $V_{14}$ |
| StandardScaler | 1.94 | 0.04 | 0.90 | 0.47 | 5.11 | 1.77 | −0.15 | 0.10 | 1.17 | 1.50 | −0.16 | 1.67 | 1.92 | 2.67 | 0.09 |
| MinMaxScaler | 0.89 | 0.01 | 0.02 | 0.01 | 0.13 | 0.10 | 0 | 0.01 | 0.04 | 0.10 | 0 | 0.13 | 0.12 | 0.12 | 0.01 |
| MaxAbsScaler | 0.89 | 0.01 | 0.02 | 0.01 | 0.13 | 0.10 | 0 | 0.01 | 0.04 | 0.10 | 0 | 0.13 | 0.12 | 0.12 | 0.01 |
| RobustScaler | 1.33 | 1.03 | 2.60 | 2.52 | 24 | 11 | 0 | 1 | 2.75 | 2.67 | 0 | 5.36 | 8.75 | 18 | 3 |
| PowerTrans | 1.83 | 0.91 | 1.56 | 1.52 | 1.94 | 1.80 | −0.25 | 2.25 | 1.50 | 1.56 | −0.22 | 1.61 | 1.69 | 1.79 | 1.73 |

result is that when all values in the dataset are nonnegative, the maximum absolute value is the maximum value, and the denominators of the two calculation methods are the same. RobustScaler scales the original feature *263* to *2.60* using the median and interquartile range (IQR) to reduce the impact of outliers on the data distribution, however some data remain unchanged. PowerTransformer can transform nonnormally distributed data into a form closer to a normal distribution, improving the distribution characteristics of the data.

Therefore, we used the Yeo-Johnson transformation in PowerTransformer [18] to standardize the representation feature vectors. Yeo-Johnson transformation can handle various forms of data, is not affected by extreme values, and allows the adjustment of parameters to find the most suitable transformation form for specific data. The Yeo-Johnson equation is shown as:

$$Y = \log(X + 1) \quad \text{if } \lambda = 0, X \geq 0 \tag{2}$$

$$Y = \frac{(X + 1)^\lambda - 1}{\lambda} \quad \text{if } \lambda \neq 0, X \geq 0 \tag{3}$$

$$Y = -\log(-X + 1) \quad \text{if } \lambda = 2, X \leq 0 \tag{4}$$

$$Y = \frac{-\left[(-X + 1)^{(2-\lambda)} - 1\right]}{2 - \lambda} \quad \text{if } \lambda \neq 2, X \leq 0 \tag{5}$$

where $X$ represents the original data, $Y$ represents the transformed data, and $\lambda$ is the transformation coefficient. The Yeo-Johnson method can be applied to automatically select an appropriate value of $\lambda$ according to the features of the data. This transformation method not only adjust the data distribution characteristics but also improve the data analyzability and stability of the model to provide higher-quality feature representation for subsequent machine learning model processing.

## 5.5 Detection model construction

Ensemble learning (EL), as an advanced strategy in machine learning, integrates multiple methods through three main techniques: bagging, stacking, and boosting. Compared with a single model, ensemble learning methods can often achieve better performance and stronger adaptability in practical applications [53]. In our research, we trained and validated 13 typical machine learning algorithms: logistic regression (LR), SVM, decision tree (DT), RF, adaptive boosting (AdaBoost), XGBoost, K-nearest neighbors (KNN), multilayer perceptron (MLP),

ridge regression (Ridge), LightGBM, CatBoost, gradient boosting decision tree (GBDT), and naive Bayes (NB). By training and validating these 13 typical machine learning algorithms, we aimed to find the optimal ensemble learning method.

To accurately identify border binaries in the firmware, we adopted stacking technology. This EL method combines the prediction results of base learners and increases the model prediction accuracy and robustness through meta-learner retraining. We found that XGBoost, LightGBM, CatBoost, and RF achieved the best prediction performance among the 13 machine learning algorithms that we trained and validated (see the corresponding test RQ2 in Sect 6.6). Therefore, we chose the XGBoost, LightGBM, and CatBoost models as the base learners and the RF model as the meta-learner to output the final XLC-R prediction.

The objective function $M_{xgb}(\Theta)$ of the base learner XGBoost consists of a loss function and a regularization term, as shown in Eq 6. Here, $y_i$ represents the true value of the $i$-th feature vector, $\hat{y}_i$ denotes the predicted value of the $i$-th feature vector, $n$ is the number of feature vectors, $L(y_i, \hat{y}_i)$ is the loss function, and $\Omega(f_k)$ is the regularization term.

$$M_{xgb}(\Theta) = \sum_{i=1}^{n} L(y_i, \hat{y}_i) + \sum_{k=1}^{K} \Omega(f_k) \tag{6}$$

Define $\Omega(f_k)$ as $\Omega(f) = \gamma T + \frac{1}{2}\lambda \sum_{j=1}^{T} w_j^2$, where $T$ is the number of leaf nodes in the tree, and $w_j$ is the weight of leaf node $j$. For the $t$-th tree, the optimal split point is determined by Eq 7, where $g_i$ and $h_i$ are the first-order and second-order gradient statistics, respectively.

$$M_{cgb}^{(t)} = \sum_{i=1}^{n} \left[ g_i w_{q(x_i)} + \frac{1}{2} h_i w_{q(x_i)}^2 \right] + \gamma T + \frac{1}{2}\lambda \sum_{j=1}^{T} w_j^2 \tag{7}$$

The objective function $M_{lgb}(\Theta)$ of the base learner LightGBM is similar to $M_{xgb}(\Theta)$, but it uses a histogram-based algorithm to find the optimal split point. The construction of the histogram is shown in Eq 8, where $H(b)$ is the value of the $b$-th histogram, and $g_i$ is the gradient value of the $i$-th feature vector.

$$H(b) = \sum_{i \in bin_b} g_i \tag{8}$$

The objective function $M_{cat}(\Theta)$ of the base learner CatBoost is also similar to $M_{xgb}(\Theta)$, but it can handle categorical features and use ordered boosting to reduce the model's sensitivity to the order of training data. CatBoost reduces gradient bias through the ordered boosting strategy. For each feature vector $i$, its gradient is calculated based on the model predictions of the previous $i-1$ feature vectors, as shown in Eq 9.

$$g_i = \frac{\partial L(y_i, \hat{y}_i^{(t-1)})}{\partial \hat{y}_i^{(t-1)}} \tag{9}$$

CatBoost uses target encoding to handle categorical features. For a categorical feature $x_j$, its encoded value is calculated as shown in Eq 10, where $I(\cdot)$ is the indicator function.

$$\text{Enc}(x_j) = \frac{\sum_{i=1}^{n} I(x_{ij} = x_j) \cdot y_i}{\sum_{i=1}^{n} I(x_{ij} = x_j)} \tag{10}$$

The meta-learner RF constructs multiple decision trees and determines the final prediction through voting. For the prediction of a single tree, given the input feature $x_i$, the predicted value of a single decision tree is shown in Eq 11, where $f_k(x_i)$ is the predicted value of the $k$-th tree.

$$\hat{y}_i^{(k)} = f_k(x_i) \tag{11}$$

RF uses Eq 12 to determine the final prediction through majority voting, where $K$ is the number of decision trees.

$$\hat{y}_i = \frac{1}{K}\sum_{k=1}^{K} f_k(x_i) \tag{12}$$

The final prediction of XLC-R can be expressed as Eq 13.

$$\hat{y}_i = M_{rf}\left(\left(M_{xgb}(x_i), M_{lgb}(x_i), M_{cat}(x_i)\right)\right) \tag{13}$$

Where $M_{\text{xgb}}(x_i)$, $M_{\text{lgb}}(x_i)$, and $M_{\text{cat}}(x_i)$ represent the predictions made by the XGBoost, LightGBM, and CatBoost models, respectively; $M_{\text{rf}}$ denotes the prediction made by the RF model; and $\hat{y}_i$ is the final prediction result. Notably, we utilized a grid search to determine the optimal settings for XLC-R during training. Please refer to Sect 6.2 for the specific parameter settings.

Algorithm 2 demonstrates the pseudocode for the stacking method applied in XLC-R. XLC-R begins by training each base learner (XGBoost, LightGBM, and CatBoost) on the training data $X_{train}$ and $y_{train}$, which generates the prediction results for the training data (train_pred) and the test data (test_pred). Then, the predictions from the base learners are stacked together to create the meta-features $X_{meta\_train}$ and $X_{meta\_test}$. Next, the meta-learner RF is trained using the meta-features $X_{meta\_train}$ and the labels $y_{train}$. Finally, the trained meta-learner is utilized to predict the meta-features $X_{meta\_test}$, yielding the final result $y_{pred}$.

**Algorithm 2.  XLC-R Stacking ensemble learning.**

```
    Input: Training data X_train, y_train; Test data X_test
    Output: Final predictions y_pred
 1  base_learners ← { # Initialize base-learners and a meta-learner
 2    `xgb': XGBoost(),
 3    `lgb': LightGBM(),
 4    `cat': CatBoost()
 5  }
 6  meta_learner ← RandomForest() # Random Forest as meta-learner
 7  base_predictions_train ← [] # Store base learners' predictions on training data
 8  base_predictions_test ← [] # Store base learners' predictions on test data
 9  for name, model ∈ base_learners.items() do
10      model.fit(X_train, y_train) # Train base learners
11      train_pred ← model.predict(X_train) # Generate predictions on training data
12      base_predictions_train.append(train_pred)
13      test_pred ← model.predict(X_test) # Generate predictions on test data
14      base_predictions_test.append(test_pred)
15  end
16  X_meta_train ← stack(base_predictions_train) # Stack base learners' predictions as
      meta-features
17  X_meta_test ← stack(base_predictions_test) # Stack base learners' predictions as
      meta-features
18  meta_learner.fit(X_meta_train, y_train) # Train the meta-learner
19  y_pred ← meta_learner.predict(X_meta_test) # Generate final predictions using meta-learner
20  return y_pred
```

# 6 Experiment and evaluation

In this section, we evaluate BBDetector by answering the following four Research Questions (RQs).

**RQ1**: Can BBDetector be used to detect border binaries, and how precise is it?

**RQ2**: Is the construction of XLC-R reasonable?

**RQ3**: Is each dimensional feature utilized in the MDFM reasonable and meaningful?

**RQ4**: How does BBDetector perform in detecting border binaries for real-world IoT firmware compared with the baseline methods?

## 6.1 Implementation

First, we developed a crawler tool for obtaining firmware based on the Scrapy framework and implemented firmware file system extraction by extending *binwalk* [49]. Then, a feature extraction tool was developed based on *Radare2* [51] to extract the behavioral and semantic features of binary executables, as well as the structure, semantics, and behavioral features of functions. Finally, the detection model was constructed via the scikit-learn [54], XGBoost [19], LightGBM [20], and CatBoost [21] libraries. The prototype system is implemented in *python*, with a total of approximately 2,580 lines of code.

## 6.2 Experimental setup

All experiments were conducted on a server with an Intel$^{®}$ Xeon(R) Silver 4214 CPU @ 2.2.GHz, 256GB RAM, 5.0TB usable HDD, and two NVIDIA V100S GPUs. The server operating system runs on the *Ubuntu 22.04 AMD64* operating system, and the runtime environment includes *Python 3.8*, *Binwalk 2.1.1*, and *Radare2 5.9.1*.

The configuration adopted by XLC-R was as follows: XGBoost used *100* estimators, with an evaluation metric of *logloss*, a learning rate of *0.1*, and a random state of *42*; LightGBM used *100* estimators, with a learning rate of *0.1*, a maximum depth of *-1* and a random state of *42*; CatBoost used *100* estimators, with a random state of *42*; and RF used *20* estimators, with a maximum depth of *4*, and a random state of *42*.

## 6.3 Dataset

**Dataset I:** We adopted the firmware collection and unpacking method described in Sect 5.1 to obtain 8,192 file systems and utilized the border binary collection method from Sect 5.2 to filter out 320 representative border binaries from 67 different device series across 10 vendors. The specific firmware details are outlined in Table 4, which displays the vendor, device series, number of device series (# Series), average number of files per firmware filesystem (Avg # File), average number of executable binaries per firmware (Avg # Bin), number of firmware models (# FirmType), number of network parsing functions (# NetFunc), and number of collected border binaries (# Border). This dataset serves as a sample set of feature vectors extracted from border and non-border binaries in real-world firmware for training and evaluating the detection model.

For the firmware listed in Table 4, we identified and collected 1,307 network parsing functions (the method is detailed in Sect 5.2. These functions are associated with 1,103 different border binaries. The remaining functions are associated with non-border binaries. Then, we extracted and constructed feature vectors for both border and non-border binaries via the MDFM. Program-level features ($V_0$ and $V_1$) can be obtained through whole binary analysis, but the function-level features differ. The function-level features of the border binaries directly correspond to the feature vectors of any network parsing functions within them.

**Table 4. Details on the firmware details used to build Dataset I.**

| Vendor | Device Series | #Series | Avg# File | Avg# Bin | #FirmType | #NetFunc | #Borber |
|--------|---------------|---------|-----------|----------|-----------|----------|---------|
| ASUS | BlueCave/BRT/AC/N/G/WL | 6 | 1,231 | 77 | 19 | 125 | 93 |
| Belkin | AC/N/F9J/F9K/F7D | 5 | 1,779 | 58 | 18 | 87 | 81 |
| D-Link | DWR/DIR/DCS | 3 | 1,673 | 99 | 49 | 214 | 201 |
| Fastcom | FD/FER/FAX/AP | 4 | 1,885 | 100 | 9 | 14 | 11 |
| Linksys | E/EA/LAPAC/LAPAN/PLW/RE/WAG/WAP/WES/WRT/WUMC/X | 12 | 1,169 | 62 | 19 | 97 | 104 |
| Netgear | AC/DST/DGN/DM/EX/JNR/JR/JWNR/N/PR/WNDR/R/WNR/XR | 14 | 1,523 | 157 | 61 | 463 | 372 |
| Tenda | AC/FH/G/W/WH | 5 | 702 | 75 | 11 | 55 | 41 |
| TP-Link | C/D/W/IPC/MR/NC/S/SIPC/WR | 9 | 984 | 129 | 58 | 134 | 101 |
| Trendnet | IP/TEG/TEW/THA/TL2/TPE/TPL | 7 | 545 | 41 | 64 | 92 | 83 |
| ZyXEL | V1/V2 | 2 | 2,201 | 149 | 12 | 26 | 16 |
| Total | —— | 67 | 13,692 | 947 | 320 | 1,307 | 1103 |

Since a border binary may have multiple parsing functions, its feature vector can match more than one function. Thus, 1,307 feature vectors were constructed from the 1,103 border binaries. The function-level feature of the non-border binaries was constructed by selecting any one of the functions contained in it, and considering the distribution of feature values of the function, 5,814 feature vectors were selected from the feature vectors of all the non-border binaries.

We used 1,307 feature vectors corresponding to the network parsing functions as positive samples, with a label of 1. The selected 5,814 feature vectors were used as negative samples, with a label of −1. This sample setup aligns with the observation that in practice, the number of border binaries is less than the number of non-border binaries. Finally, we constructed Dataset I, which contains 7,121 feature vectors.

**Dataset II:** A set of 49 firmware images based on Linux OS and sourced from Karonte [11] was used as Dataset II. This dataset was utilized for comparison with the baseline methods Karonte and SaTC. Table 5 provides detailed information in Dataset II, including the vendor, device series, number of firmware images (# Firm), architecture (Arch), number of files in the file system (# File), number of executable binaries (# Bin), and average number of border binaries per executable binary (Avg # Border).

## 6.4 Baseline

We chose Karonte [11] and SaTC [12] as the baseline methods. Karonte represents the first method for identifying border binaries through network parsing functions, whereas SaTC identifies border binaries via front-end string matching. These two methods represent two different technical routes. Furthermore, Karonte and SaTC are commonly used as baselines in border binary detection research [4,10,14,24]. Therefore, we also chose Karonte and SaTC as baseline methods for comparison.

**Table 5. Dataset II.**

| Vendor | Device Series | #Firm | Arch | #File | #Bin | Avg #Borber |
|--------|---------------|-------|------|-------|------|-------------|
| D-Link | DIR/DWR/DCS | 9 | ARMel/MIPSel/MIPSeb | 15,646 | 1,884 | 6 |
| Netgear | R/XR/WNR | 17 | ARMel/MIPSel | 33,664 | 5,324 | 7 |
| TP-Link | TD/WA/WR/TX/KC | 16 | ARMel/MIPSel/MIPSel | 11,180 | 2,340 | 5 |
| Tenda | AC/WH/FH | 7 | ARMel/MIPSel | 4,410 | 1,421 | 5 |
| Total | —— | 49 | —— | 64,900 | 10,969 | 6 |

## 6.5 Evaluation metrics

We evaluated BBDetector via five indicators: precision ($Pre = \frac{TP}{TP+FP}$), recall ($R = \frac{TP}{TP+FN}$), F1 score ($F1 = \frac{2 \cdot \text{Precision} \cdot \text{Recall}}{\text{Precision} + \text{Recall}}$), receiver operating characteristic (ROC), and the area under the ROC curve (AUC), where TN, FN, TP, and FP are the numbers of true-negative, false-negative, true-positive, and false-positive samples, respectively. The reason we chose these five metrics is as follows:

Precision (*Pre*) refers to the ratio of the number of border binaries identified correctly to the total number of instances predicted as border binaries. A higher precision indicates that the model is more accurate in its predictions regarding border binaries, resulting in a reduced likelihood of false positives. Reducing false positive rates is crucial in security analysis since incorrect predictions can lead to unnecessary resource consumption. Thus, precision is a key metric for evaluating the effectiveness of border binary detection methods.

Recall (*R*) refers to the proportion of border binaries correctly identified by the model among all border binaries. The higher the recall is, the stronger the performance of the detection model, and the lower the likelihood of false negatives. This is essential for ensuring comprehensive security analysis of border binaries.

F1 score (*F1*) is the weighted average of precision and recall, providing a balanced metric for evaluating the overall performance of a model. Balancing precision and recall is crucial in practical applications, as a model with high precision but low recall may overlook important cases, while a model with high recall but low precision may produce a significant number of false positives.

The ROC curve is the receiver operating characteristic curve. The closer the ROC curve is to the coordinate (0,1), the better the performance of the learner. The AUC value is the area under the ROC curve, and higher AUC values, the stronger the model's ability to distinguish between border and non-border binaries.

## 6.6 Experimental results and discussion

**RQ1: Effectiveness of BBDetector.** To verify whether BBDetector can be used to detect border binaries, we trained and validated it in Dataset I. To ensure the effectiveness of the training process and the rationality of the validation process, we selected 80% of the border and non-border binary samples as the training sample set and the remaining 20% of the samples as the validation sample set.

In the training phase, the training sample set is used independently to train three base learners (XGBoost, LightGBM, and CatBoost). After training, each base learner generates its prediction results, which are then combined into a new feature vector and fed into a meta learner (RF) for training, resulting in the final XLC-R model. The goal of the meta-learner is to learn how to effectively integrate the predictions of the three base learners to form a more robust detection model, as described in Sect 5.5. The rationality analysis of the construction for XLC-R is presented in response to RQ2.

To illustrate the improved detection effect of XLC-R over those of the individual base learners, we also evaluated the standalone performance of the base learners. The precision, recall, and F1 score results for the three base learners and XLC-R on the validation set are shown in Table 6. The ROC curve and AUC values are shown in Fig 3.

Table 6 shows that the precision, recall, and F1 score results for XLC-R were 94.98%, 91.02%, and 92.84%, respectively. The results of XLC-R exceed those of the three base learning methods. Therefore, XLC-R substantially outperformed the three base learners in classification. This result is attributed to the stacking method's ability to reasonably integrate the prediction results of each base learner to achieve better performance.

**Table 6. Precision, recall, and F1 score results of the base learners and XLC-R on the validation set.**

| Metrics | XGBoost | LightGBM | Catboost | XLC-R |
|---|---|---|---|---|
| **Pre(%)** | 93.17 | 91.70 | 92.18 | 94.98 |
| **R(%)** | 85.93 | 85.92 | 82.96 | 91.02 |
| **F1(%)** | 89.40 | 88.72 | 87.72 | 92.84 |

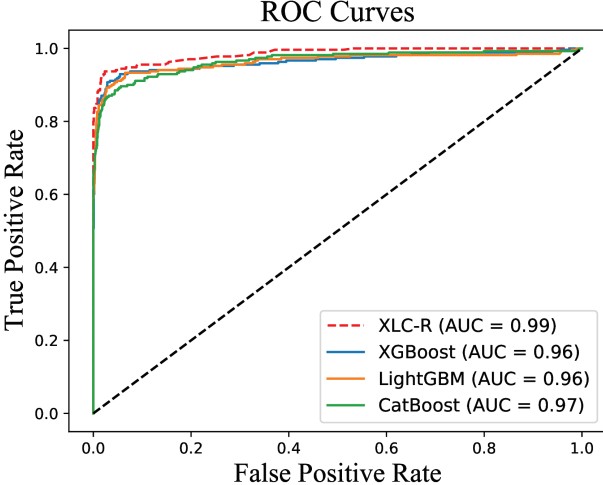

**Fig 3. ROC curves of the three base learners and XLC-R on the validation set.**

Fig 3 shows that the ROC curve for XLC-R was closer to the coordinate (0, 1), and that AUC value was 0.99. The AUC values for XGBoost, LightGBM, and CatBoost were 0.96, 0.96, and 0.97, respectively. XLC-R has the highest AUC value, indicating it possesses the strongest ability to distinguish between border and non-border binaries.

The above tests reveal that BBDetector can effectively detect border binaries, with high precision, recall, F1 score, and AUC values.

**RQ2: Detection model rationality.** BBDetector uses a stacking method for border binary detection, leveraging model complementarity to enhance the recognition of intricate data patterns. To construct effective stacking models, we assessed 13 known machine learning algorithms (Sect 5.5) in terms of classification efficacy.

In this test, 80% of the border and non-border binaries from Dataset I were allocated to the training set, with the remaining binaries used for validation. Using the default configuration, the precision, recall, and F1 score values of the 13 machine-learning models on the validation set are shown in Fig 4.

The result reveals that RF, XGBoost, LightGBM, and CatBoost obtain the highest precision and F1 score values among the 13 machine learning models tested. The F1 score values were 86.29%, 89.40%, 88.72%, and 87.33%, respectively. Importantly, the F1 score values for XGBoost, LightGBM, and CatBoost surpass that of RF.

Since the performance of each base classifier directly influences the performance of the meta-classifier, the classification results of the meta-learner are usually better when the classification results of the base classifier are superior classification outcomes. Consequently, the final detection model was constructed based on RF, XGBoost, LightGBM, and CatBoost. XGBoost, LightGBM, and CatBoost, which had the highest F1 scores, were selected as the

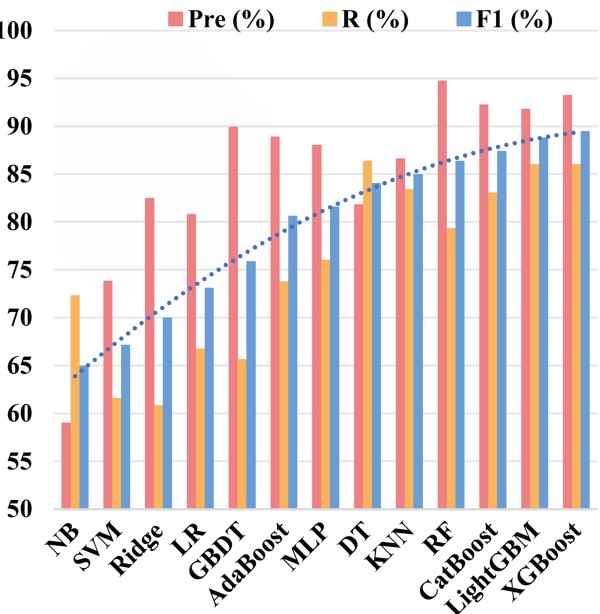

**Fig 4. Precision, recall, and F1 score values of 13 established machine learning models on the validation set.**

base learners, whereas RF served as the meta-learner. The effectiveness of XLC-R has been validated in response to RQ1.

**RQ3: Feature selection rationality.** The LASSO regression method was used to evaluate the rationality of the 15-dimensional features in MDFM. LASSO regression can automatically identify important features without manual intervention or complex feature selection procedures. Feature selection is achieved by effectively shrinking the coefficients of less important features to zero [55].

We used the LassoCV class in the scikit-learn library, a LASSO regression method with cross-validation, to screen and test the initial 15-dimensional features. Regarding the parameter configuration, we implemented 10-fold cross-validation to optimize the regularization parameter *alpha*, and set the random state to *42* for obtaining more reliable analysis results.

The inputs of the meta-learner are the prediction results of the base learners rather than a 15-dimensional feature vector. That is, the feature selection process is only used directly in the training processes of the three base learners (XGBoost, LightGBM, and CatBoost). Therefore, we trained and validated the base learners. For Dataset I, the test results are shown in Fig 5.

The results show that none of the 15-dimensional features were eliminated, indicating the significance of each dimensional feature in the MDFM for border binary detection. In XGBoost, the feature weights assigned to $V_3$, $V_8$, $V_{10}$, and $V_{13}$ are quite small, whereas $V_7$ and $V_{12}$ contribute more significantly. In LightGBM, the feature weights assigned to $V_6$, $V_{10}$, and $V_{13}$ are relatively small, while $V_1$, $V_2$, $V_3$, $V_7$, and $V_{12}$ have relatively larger weights. In CatBoost, the feature weights assigned to $V_3$, $V_6$, $V_{10}$, and $V_{13}$ are relatively small, while $V_1$, $V_7$, and $V_{12}$ carry relatively larger weights.

The feature weights reflect their effects on model predictions, with higher weights indicating greater influence. The different feature weights reveal that each dimensional feature of the MDFM contributes uniquely to different models. For example, APIs related to string comparison and strings related to HTTP requests contribute significantly, as they are crucial for

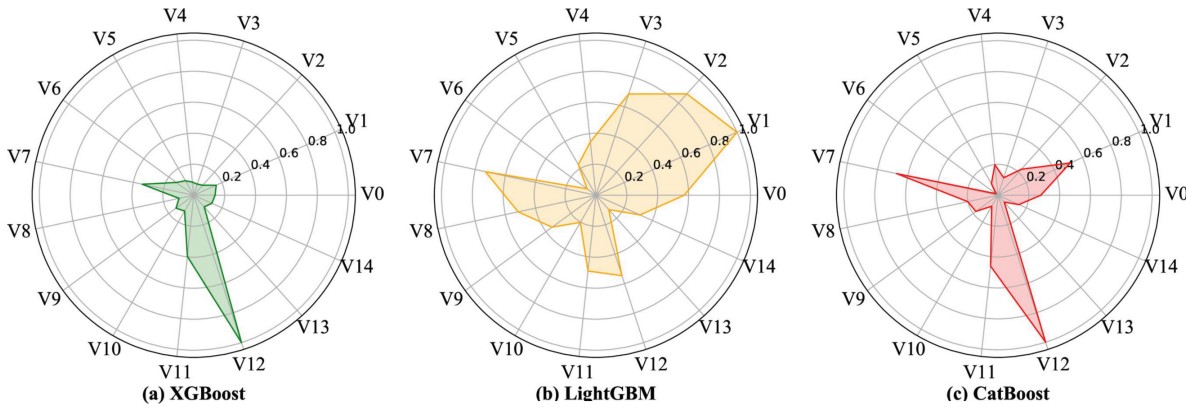

**Fig 5. Feature importance measurements for XGBoost, LightGBM, and CatBoost.**

validating request data through network parsing functions in border binaries. In contrast, the contributions of APIs related to string conversion and strings related to MQTT requests are relatively small, probably because the string conversion-related APIs are used only for auxiliary functionality in the network parsing function, and MQTT request-related strings are involved only in establishing connections rather than in the main part of actually processing the network input. In short, these 15-dimensional features of the MDFM are reasonable.

**RQ4: Comparison with existing methods.** To further evaluate the advantages of BBDetector, we compared it to the baseline method in Sect 6.4.

Table 7 shows the detection results of Karonte, SaTC, and BBDetector in Dataset II, along with detailed information regarding the detected firmware, including the device vendor, number of firmware samples (#Firm), number of files in the firmware (#File), number of binaries (#Bin), total number of instances predicted as border binaries (#P), number of border binaries identified correctly (#TP), the ratio of the number of border binaries identified correctly to the total number of instances predicted as border binaries (*Pre*), and the fraction of the total border binaries that the model can correctly identify as border binaries (*R*). Because it was not entirely certain whether all border binaries in the firmware were detected, the set of border binaries detected by the three methods was treated as the actual set of border binaries for the corresponding firmware.

The results show that Karonte, SaTC, and BBDetector identified 57, 83, and 185 border binaries in Dataset II, respectively. BBDetector detected 3.25 times and 2.23 times more border binaries than Karonte and SaTC did, respectively. Moreover, the precision rates of Karonte, SaTC, and BBDetector for 49 firmware were 31.32%, 56.46%, and 57.81%, respectively, and the recall rates were 25.11%, 36.89%, and 81.50%, respectively. Obviously, the

**Table 7. Details on border binaries detected by Karonte, SaTC, and BBDetector in Dataset II.**

| Vendor | #Firm | #File | #Bin | Karonte | | | | SaTC | | | | BBDetector | | | |
|---|---|---|---|---|---|---|---|---|---|---|---|---|---|---|---|
| | | | | #P | #TP | Pre(%) | R(%) | #P | #TP | Pre(%) | R(%) | #P | #TP | Pre(%) | R(%) |
| D-Link | 9 | 15,646 | 1,884 | 55 | 31 | 56.36 | 54.39 | 27 | 10 | 37.04 | 17.54 | 72 | 39 | 54.17 | 68.84 |
| Netgear | 17 | 33,664 | 5,324 | 71 | 17 | 23.94 | 15.89 | 51 | 43 | 84.31 | 40.19 | 157 | 92 | 58.60 | 85.98 |
| TP-Link | 16 | 11,180 | 2,340 | 23 | 4 | 17.39 | 9.09 | 48 | 21 | 43.75 | 43.75 | 64 | 38 | 59.38 | 86.37 |
| Tenda | 7 | 4,410 | 1,421 | 33 | 5 | 15.15 | 26.16 | 21 | 9 | 42.86 | 47.37 | 27 | 16 | 59.26 | 84.21 |
| **Total** | **49** | **64,900** | **10,969** | **182** | **57** | **31.32** | **25.11** | **147** | **83** | **56.46** | **36.89** | **320** | **185** | **57.81** | **81.50** |

precision and recall of BBDetector were the highest. Therefore, BBDetector outperformed Karonte and SaTC in border binary detection.

In Fig 6, we visually compare border binaries detected by Karonte, SaTC, and BBDetector in Dataset II.

As shown in Fig 6, Karonte identified 31 border binaries in D-Link firmware images, 17 in Netgear firmware images, 4 in TP-Link firmware images, and 5 in Tenda firmware images. SaTC identified 10 border binaries in D-Link firmware images, 43 in Netgear firmware images, 21 in TP-Link firmware images, and 9 in Tenda firmware images. BBDetector identified 39 border binaries in D-Link firmware images, 92 in Netgear firmware images, 38 in TP-Link firmware images, and 16 in Tenda firmware images. BBDetector successfully detected the majority of the border binaries found by both Karonte and SaTC. Taking 9 firmware samples of *D-Link* as an example, compared with Karonte, BBDetector detected 22 border binaries that were detected by Karonte and an additional 17 more border binaries than Karonte. Compared with SaTC, BBDetector detected 8 border binaries detected by SaTC and an additional 31 border binaries. Overall, BBDetector successfully detected 9 border binaries that Karonte and SaTC could not detect.

However, BBDetector outputted some false negatives for the 9 firmware of *D-Link*, compared with Karonte and SaTC. Karonte detected 9 border binaries that BBDetector missed. This discrepancy may be because Karonte characterizes and detects border binaries only based on the HTTP protocol, whereas BBDetector covers multiple protocol features, which may weaken the feature characterization ability of HTTP protocol-related border binaries, resulting in a lower recognition performance than that of Karonte for HTTP protocol-related border binaries. SaTC identified 2 border binaries that BBDetector missed, possibly because SaTC detects shared strings between the front and back ends, covering all strings in the front end of the firmware. BBDetector uses protocol universal string features, so it is not sensitive to personalized string features, but this design helps ensure its effectiveness.

Table 8 shows the detection results for 8 representative firmware samples, where #TP represents the true number of border binaries and Border Set represents the set of border binaries.

As shown in the table, Karonte identified 11 border binaries across 8 firmware images, SaTC identified 13 border binaries, and BBDetector identified 28 border binaries. BBDetector detected 15 more border binaries than SaTC. Notably, Karonte did not detect any border binaries in the D-Link DIR-826L or Tenda WH450 firmware because it is limited by the binary analysis tool angr [56] and cannot analyze some MIPS architecture binaries. In addition, SaTC had a high false-negative rate, mainly because it outputs only the top three binary executables for each firmware as border binaries. In contrast, BBDetector exhibited better

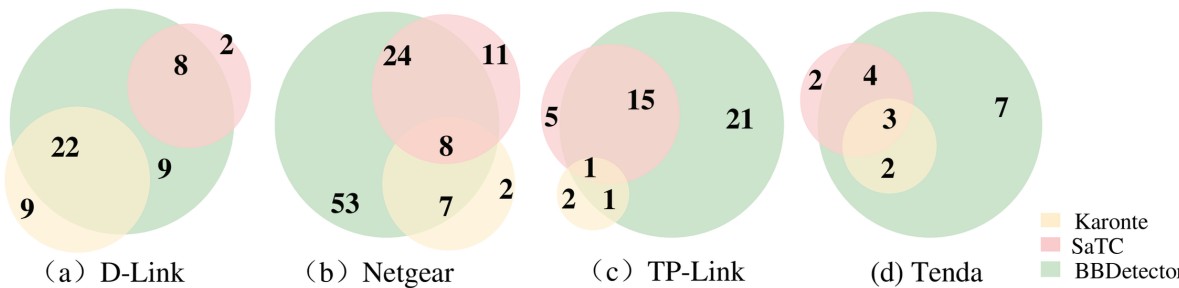

**Fig 6. Comparison of border binaries detected by BBDetector and the baseline methods in Dataset II.**

**Table 8. Detailed information regarding the border binaries obtained by BBDetector and baseline methods for 8 representative firmware samples.**

| Vendor | Firmware | #TP | Karonte | | SaTC | | BBDetector | |
|---|---|---|---|---|---|---|---|---|
| | | | #TP | Border Set | #TP | Border Set | #TP | Border Set |
| D-Link | DIR-826L | 4 | null | null | 2 | ncc, minidlna | 3 | ncc, signalc, miniupnpd |
| D-Link | DIR-880L | 7 | 6 | httpd, minidlna, rgbin,wifidog, proxyd, cgibin | 0 | 0 | 5 | minidlna, wifidog,httpd, gdataclient, rgbin |
| Netgear | R6200 | 7 | 1 | ookla | 3 | httpd, acos_service, upnpd | 5 | bftpd, minidlna.exe,pppd, httpd, upnpd |
| Netgear | R7900 | 7 | 1 | httpd | 3 | httpd, upnpd, acos_service | 6 | curl, httpd, forked-daapd,pppd, minidlna.exe, upnpd |
| TP-Link | C3200 | 5 | 3 | minidlnad, httpd, cwmp | 2 | minidlnad, pppd | 4 | rttd, minidlnad, cwmp, pppd |
| TP-Link | C5 | 2 | 0 | 0 | 1 | httpd | 2 | httpd, minidlnad |
| Tenda | WH450 | 1 | null | null | 1 | httpd | 1 | httpd |
| Tenda | AC6 | 3 | 0 | 0 | 1 | pppd | 2 | miniupnpd, dhttpd |
| Total | 8 | 36 | 11 | —— | 13 | —— | 28 | —— |

performance in detecting border binaries. For example, in the *D-Link DIR-826L* firmware, Karonte did not identify any border binary, SaTC identified 2 border binaries, and BBDetector identified 3 border binaries.

The superior detection capability of BBDetector stems from two pivotal factors. First, the open-source binary analysis tool *Radare2* [51] has cross-architecture analysis capabilities, effectively supporting various architectures. Second, the MDFM effectively characterizes the key features of border binaries, and the stacking method has strong generalizability through training.

BBDetector provides important guidance and support for firmware security analysis. Replacing the border binaries and network parsing functions identified by Karonte and SaTC with the border binaries and network parsing functions detected by BBDetector and performing subsequent security analysis of Karonte and SaTC, firmware vulnerabilities, as shown in Table 9, can be successfully found. SaTC failed to detect the *minidlna* border binary in the *DIR-880LA1-FW107WWb08* firmware and did not detect three security vulnerabilities, namely, CVE-203-39669, CVE-2023-39671, and CVE-2023-39674, in *minidlna*. Similarly, Karonte failed to detect the ncc border binary in the *DIR-826LA1_FW105B13* firmware, resulting in the missed detection of CVE-2021-45382.

**Table 9. List of known vulnerabilities found in Dataset II by BBDetector.**

| Vendor | Firmware | Bug IDs |
|---|---|---|
| D-Link | DIR-880LA1-FW107WWb08 | CVE-2023-39669,CVE-2023-39671, CVE-2023-39674 |
| D-Link | DIR-826LA1_FW105B13 | CVE-2021-45382 |
| Netgear | R7900-V1.0.1.26_10.0.23 | CVE-2020-28373,CVE-2021-45540,CVE-2021-45606 |
| Netgear | WNR3500Lv2-V1.2.0.46 | CVE-2021-45525 |
| Netgear | XR500-V2.1.0.4 | CVE-2020-26913,CVE-2021-29069,CVE-2021-45623 |
| Netgear | R6400v2-V1.0.2.46_1.0.36 | CVE-2021-45525,CVE-2021-45606,CVE-2023-36187 |
| Netgear | R6700-V1.0.1.36_10.0.40 | CVE-2021-45525 |
| Netgear | R7000P-V1.3.0.8_1.0.93 | CVE-2020-28373,CVE-2021-45525,CVE-2021-45606,CVE-2021-38521,CVE-2023-36187 |
| Netgear | R8000-V1.0.4.4_1.1.42 | CVE-2021-45525,CVE-2021-45539,CVE-2021-45540 |
| Netgear | R8900-V1.0.2.40 | CVE-2020-26913 |

## 7 Discussion

Although BBDetector outperforms the current state-of-the-art methods in border binary detection, several limitations remain.

**The MDFM needs further optimization.** Compared with existing methods, BBDetector detected more border binaries; however, there were also several false negatives. We believe that optimizing the MDFM can further improve the detection accuracy.

**The limited number of border binaries restricts the training level of the detection model.** Owing to the current lack of large-scale datasets for border binaries, we spent considerable time and manpower screening and labeling border binaries and ultimately found only 1,103 different border binaries in 8,192 firmware file systems.

## 8 Conclusion and future work

To address the insufficient detection ability problems of the existing border binary detection methods, we propose BBDetector, a border binary detection method based on the MDFM. The MDFM covers the behavioral and semantic features of binaries, as well as the structural, behavioral, and semantic features of functions, to characterize the border binary features. BBDetector uses XGBoost, LightGBM, and CatBoost as base learners and RF as a meta-learner, and trains on the sample sets of the border and non-border binaries that we constructed for the first time to obtain XLC-R. The results show that the AUC value of XLC-R was 0.99 in Dataset I. Meanwhile, BBDetector detected more border binaries than Karonte and SaTC did in Dataset II. In summary, BBDetector significantly improves the detection performance of border binaries in IoT firmware, providing important reference values for firmware security analysis.

In future studies, we aim to optimize existing methods. First, we plan to expand the MDFM by incorporating additional important features while examining the impact of the current 15-dimensional features on false negatives and eliminating feature dimensions that cause high interference with the detection results. Second, we intend to collect more border binaries through the application of BBDetector to enrich our sample set and continually optimize the existing detection models via iterative training methods to improve the detection accuracy of border binaries. Third, we explore diversified techniques, such as privacy prediction [57], signal acquisition and processing [58–60], to open up new research pathways for enhancing the accuracy of border binary detection.

## Author contributions

**Conceptualization:** Weihua Jiao.

**Data curation:** Guimin Zhang.

**Formal analysis:** Shudan Yue.

**Funding acquisition:** Guimin Zhang.

**Investigation:** Shudan Yue, Xiaonan Li.

**Methodology:** Shudan Yue.

**Resources:** Qingbao Li.

**Software:** Wenbo Zhang.

**Supervision:** Qingbao Li.

**Validation:** Shudan Yue.

**Visualization:** Wenbo Zhang.

**Writing – original draft:** Shudan Yue, Guimin Zhang.

**Writing – review & editing:** Qingbao Li, Xiaonan Li, Weihua Jiao.

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
