## [Decision Letter · Decision Letter 0]

19 Feb 2025

PONE-D-24-50757BBDetector: Intelligent Border Binary Detection in IoT Device Firmware Based on a Multidimensional Feature ModelPLOS ONE

Dear Dr. Zhang,

Thank you for submitting your manuscript to PLOS ONE. After careful consideration, we feel that it has merit but does not fully meet PLOS ONE’s publication criteria as it currently stands. Therefore, we invite you to submit a revised version of the manuscript that addresses the points raised during the review process.

We look forward to receiving your revised manuscript.

Kind regards,

Fredrick Romanus Ishengoma

Academic Editor

PLOS ONE

**Journal Requirements:**

1. When submitting your revision, we need you to address these additional requirements.Please ensure that your manuscript meets PLOS ONE's style requirements, including those for file naming. The PLOS ONE style templates can be found at https://journals.plos.org/plosone/s/file?id=wjVg/PLOSOne_formatting_sample_main_body.pdf and https://journals.plos.org/plosone/s/file?id=ba62/PLOSOne_formatting_sample_title_authors_affiliations.pdf 2. Please note that PLOS ONE has specific guidelines on code sharing for submissions in which author-generated code underpins the findings in the manuscript. In these cases, we expect all author-generated code to be made available without restrictions upon publication of the work. Please review our guidelines at https://journals.plos.org/plosone/s/materials-and-software-sharing#loc-sharing-code and ensure that your code is shared in a way that follows best practice and facilitates reproducibility and reuse. 3. Thank you for stating in your Funding Statement: This research was supported by the fund of Laboratory for Advanced Computing and Intelligence Engineering [grant number 2023-LYJJ-01-032], and the National Key Research and Development Program of China [grant number 2021YFB3101804]. Please provide an amended statement that declares *all* the funding or sources of support (whether external or internal to your organization) received during this study, as detailed online in our guide for authors at http://journals.plos.org/plosone/s/submit-now. Please also include the statement “There was no additional external funding received for this study.” in your updated Funding Statement.  Please include your amended Funding Statement within your cover letter. We will change the online submission form on your behalf. 4. Thank you for stating the following in the Acknowledgments Section of your manuscript: This research was supported by the fund of Laboratory for Advanced Computing and 680 Intelligence Engineering [grant number 2023-LYJJ-01-032], and the National Key 681 Research and Development Program of China [grant number 2021YFB3101804]. We note that you have provided funding information that is not currently declared in your Funding Statement. However, funding information should not appear in the Acknowledgments section or other areas of your manuscript. We will only publish funding information present in the Funding Statement section of the online submission form. Please remove any funding-related text from the manuscript and let us know how you would like to update your Funding Statement. Currently, your Funding Statement reads as follows:  This research was supported by the fund of Laboratory for Advanced Computing and Intelligence Engineering [grant number 2023-LYJJ-01-032], and the National Key Research and Development Program of China [grant number 2021YFB3101804]. Please include your amended statements within your cover letter; we will change the online submission form on your behalf. 5. When completing the data availability statement of the submission form, you indicated that you will make your data available on acceptance. We strongly recommend all authors decide on a data sharing plan before acceptance, as the process can be lengthy and hold up publication timelines. Please note that, though access restrictions are acceptable now, your entire data will need to be made freely accessible if your manuscript is accepted for publication. This policy applies to all data except where public deposition would breach compliance with the protocol approved by your research ethics board. If you are unable to adhere to our open data policy, please kindly revise your statement to explain your reasoning and we will seek the editor's input on an exemption. Please be assured that, once you have provided your new statement, the assessment of your exemption will not hold up the peer review process. 6. Please amend either the title on the online submission form (via Edit Submission) or the title in the manuscript so that they are identical.

Reviewers' comments:

Reviewer's Responses to Questions

**Comments to the Author**

1. Is the manuscript technically sound, and do the data support the conclusions?

Reviewer #1: Yes

Reviewer #2: Yes

2. Has the statistical analysis been performed appropriately and rigorously? 

Reviewer #1: Yes

Reviewer #2: Yes

3. Have the authors made all data underlying the findings in their manuscript fully available?

Reviewer #1: Yes

Reviewer #2: Yes

4. Is the manuscript presented in an intelligible fashion and written in standard English?

Reviewer #1: Yes

Reviewer #2: Yes

5. Review Comments to the Author

**Reviewer #1: **This paper introduces a border detection scheme based on a multidimensional feature model, presenting several interesting results. Overall, the paper is well-organized and well-written. However, I have the following comments and suggestions for improvement:

1.The motivation behind this study needs to be further clarified. Specifically, the definition and importance of the border binary should be elaborated.

2. While the authors identify three challenges in detecting border binaries, they do not adequately explain why existing state-of-the-art solutions fail to overcome these challenges.

3. Considering that the detection model training consumes considerable energy, it is important to discuss the energy consumption of nodes within IoT networks. Are these nodes active, such as LoRa nodes as discussed in “Impact of LoRa Imperfect Orthogonality: Analysis of Link-Level Performance”, or passive, like RFID nodes in “Revisiting RFID Missing Tag Identification: Theoretical Foundation and Algorithm Design”? Clarifying this critical concern and referencing these studies would enhance the paper’s practicality and relevance.

4.An analysis of the time complexity of Algorithm 1 is required.

5. The relationship between the multimodal features requires further elaboration. Are these features complementary, enhancing the overall detection accuracy, or redundant?

**Reviewer #2:** Reviewer Comments on BBDetector: Intelligent Border Binary Detection in IoT Device Firmware Based on a Multi-Dimensional Feature Model

1. The abstract should provide a concise summary of the research problem, methods, results, and significance. Currently, it lacks clarity on the specific contributions of the study.

2. Expand the literature review to include more recent studies on Border Binary detection in IoT device. Highlight how this proposed research work differentiates from existing research.

3. Provide a detailed explanation of the ensemble learning, combining extreme gradient boosting, light gradient boosting machine, and categorical boosting as base learners with random forest as the meta-learner. Include mathematical formulations and pseudocode for better understanding.

4. Authors should include the hyperparameters of the Gradient boosting model, light gradient boosting machine.

5. Authors should also include the hyperparameters of the light gradient boosting machine.

6. Elaborate on the rationale behind the choice of features selection via MDFM. Explain the feature selection process in detail.

7. The authors should justify why they chose the specific evaluation metrics used in the study. They should also provide a baseline for comparison with other methods.

8. Provide a comprehensive description of the datasets used, including their sources, characteristics, and any preprocessing steps taken before model training.

9. Authors should describe the experimental setup, including the hardware and software configurations, to ensure reproducibility of the results.

10. Incorporating relevant and recent academic sources could strengthen your paper’s validity and give readers more context and background. Some researches listed below which studied similar problems can be discussed: a).GA-mADAM-IIoT: A new lightweight threats detection in the industrial IoT via genetic algorithm with attention mechanism and LSTM on multivariate time series sensor data. b). A novel hybrid autoencoder and modified particle swarm optimization feature selection for intrusion detection in the internet of things network. c). Modified genetic algorithm and fine-tuned long short-term memory network for intrusion detection in the internet of things networks with edge capabilities.

11. A concluding paragraph explaining the research gap is needed in Section 2 related works. A more in-depth comparison highlighting how these works fall short in addressing the challenges the proposed work aims to solve would be beneficial in the end of related work section.

12. Please Change the “conclusion” section to “Conclusion and Future Work” and write future work.

13. Numerical results are good enough, but more explanations are required to analyze each figure presented.

14. Proofread the manuscript carefully to eliminate any grammatical errors or typos and ensure clarity and coherence in writing. Additionally, adhere to the formatting and style guidelines specified by the journal to enhance the professionalism of the manuscript.

6. PLOS authors have the option to publish the peer review history of their article (what does this mean?). If published, this will include your full peer review and any attached files.

Reviewer #1: No

Reviewer #2: **Yes: **OLAREWAJU "OLA" RAJI and Yakub Kayode Saheed

---

## [Author Response · Author response to Decision Letter 1]

11 Mar 2025

Dear Reviewers,

Thank you for your time and effort in reviewing our work and constructive suggestions on our manuscript. Based on your comments, we have carefully revised the manuscript to address all the issues raised. All revisions have been highlighted in yellow in a separate file labeled ‘Revised Manuscript with Track Changes’. Below, we provide a point-by-point response to your comments and describe the changes made to the manuscript.

Reviewer #1

This paper introduces a border detection scheme based on a multidimensional feature model, presenting several interesting results. Overall, the paper is well-organized and well-written.

Comment 1: The motivation behind this study needs to be further clarified. Specifically, the definition and importance of the border binary should be elaborated.

Response: We sincerely appreciate your valuable feedback on the definition and importance of border binaries. Upon reflection, we recognize that our initial explanation of border binaries was insufficient due to oversight, which may have hindered a clear understanding of their significance and the motivation behind this study. To address this, we have revised the manuscript to provide a more detailed explanation of the definition and importance of border binaries, as well as the motivation for this research. The relevant revisions can be found on Pages 5-6, Lines 202–215.

Revision text:

Border binaries are the executable binaries within IoT firmware that directly interact with external network data. They are responsible for receiving, parsing, and processing network requests from external sources, then passing the results to other modules within the firmware for further processing. As critical components of firmware, the importance of border binaries is reflected in two key aspects. First, they are involved not only in fundamental network communication operations, such as receiving and sending data, but also in handling critical tasks like identity authentication and firmware updates, playing a vital role in the functionality of the firmware. Second, due to the widespread vulnerabilities inherent in border binaries, they have become the primary entry point for most malicious data aiming to infiltrate the firmware, posing a significant threat to firmware security. Therefore, accurately identifying border binaries is crucial for ensuring the security of IoT devices, as the vulnerabilities in these binaries could allow attackers to breach the device’s defenses, infiltrate the firmware, and severely compromise the security of the entire IoT system.

Comment 2: While the authors identify three challenges in detecting border binaries, they do not adequately explain why existing state-of-the-art solutions fail to overcome these challenges.

Response: We thank the reviewer for raising this issue. Based on the reviewer’s suggestion, we have added a detailed explanation in the manuscript of why existing solutions fail to overcome these challenges. The relevant revisions can be found on Pages 2-3, Lines 44–68.

Revision text:

C1: Selection of Border Binary Features. Different IoT devices may have different border binaries, which vary depending on the network protocol used. Existing solutions, such as Karonte [12] and SaTC [13], exhibit notable limitations in feature selection: Karonte [12] utilizes only five features for detecting border binaries, while SaTC [13] relies on shared keywords between the web front-end and back-end. These methods do not fully capture the essential features of the border binaries, leading to inaccurate detection results. To date, there is no clear standard to guide feature selection and extraction for border binaries. Therefore, determining which features can accurately and comprehensively characterize the border binaries is the primary challenge in achieving accurate detection.

C2: Construction of a Border Binary Dataset. Constructing a high-quality border binary dataset is crucial for improving the robustness of border binary detection models. Existing solutions [14-16] either rely on heuristic methods to identify border binaries without using datasets, or rely on small-scale datasets, failing to allocate sufficient resources to build widely used border binary sets. Given the current lack of open-source border binary datasets for training machine learning (ML) models, it is necessary to construct such a dataset through automated tools and manual review. However, this part of the work heavily relies on expert experience and requires significant human resources.

C3: Choice of a Border Binary Detection Model. Existing solutions [9,12,17] perform poorly in identifying diverse border binaries. For example, rule-based methods [12,17] struggle to adapt to the protocols of diversified devices, while traditional machine learning-based solutions [16] exhibit limited generalization ability when encountering unseen border binaries. Therefore, selecting an appropriate border binary detection model is crucial for achieving higher accuracy and improving the efficiency of border binary detection.

Comment 3: Considering that the detection model training consumes considerable energy, it is important to discuss the energy consumption of nodes within IoT networks. Are these nodes active, such as LoRa nodes as discussed in “Impact of LoRa Imperfect Orthogonality: Analysis of Link-Level Performance”, or passive, like RFID nodes in “Revisiting RFID Missing Tag Identification: Theoretical Foundation and Algorithm Design”? Clarifying this critical concern and referencing these studies would enhance the paper’s practicality and relevance.

Response: We sincerely thank the reviewer for their valuable suggestions. Inspired by the recommended literature, we have supplemented the related work with a method for locating border binaries through dynamic emulation of protocol services that receive external network inputs in firmware, categorizing it as the third category of border binary detection method. However, given the high computational resource requirements of dynamic emulation and the inherent challenges of firmware emulation, this paper primarily focuses on static heuristic-based methods rather than dynamic emulation approaches. Although training the detection model consumes some energy, our model has lower energy consumption during the detection phase and avoids the additional resource demands of dynamic emulation.

Furthermore, our current detection model mainly targets the code processing logic in the static environment of firmware and does not specifically distinguish between active and passive characteristics of nodes. While dynamic emulation methods may uncover more hidden border binaries, their implementation complexity and resource consumption are significant, and thus, we have listed them as a direction for future research. We have referenced the studies mentioned by the reviewer (references [26] and [28]) in the relevant section to enhance the practicality and relevance of our work. The specific revisions are on Page 4, Lines 116-120.

Revision text:

An analysis of existing research shows that methods for identifying critical binaries in IoT firmware can be divided into three categories. The first category consists of approaches that rely on expert knowledge, where researchers manually select potential critical binaries (such as httpd and cgibin) as objects for security analysis [8-11,15,23-25]. The second category consists of approaches that are based on heuristic methods, automatically identifying critical firmware binaries as objects through feature analysis [12-15]. The third category consists of approaches that dynamically emulate the external communication services [26-29] received in IoT firmware. However, the first category of approaches relies too heavily on expert knowledge, while the third category of approaches requires device emulation, which demands significant computational resources and poses challenges in firmware emulation. This paper primarily focuses on the second category of method, which involves automated identification.

Comment 4: An analysis of the time complexity of Algorithm 1 is required.

Response: Thanks for your suggestion regarding the time complexity analysis of Algorithm 1. We have revised the manuscript to include the time complexity analysis of Algorithm 1. These changes can be found on Page 12, Lines 451–461.

Revision text:

Suppose that binary set B contains the number of b is n, the average number of functions in each binary b is m, and k is the average number of feature extraction operations in each function f. The time complexity of Algorithm 1 mainly consists of the following parts: (1) outer loop: traversing each binary b in binary set B, with time complexity of O (n); (2) Inner loop: Traversing each function f in each binary b, with time complexity of O (m); (3) Internal operation: performing feature extraction on each function f, including counting the number of basic blocks, branch statements, API calls, and string matches. Assuming the average number of these operations in each function f is a constant k, the time complexity of the internal operations is O (k). Therefore, the total time complexity of Algorithm 1 is O (n × m × k).

Comment 5: The relationship between the multimodal features requires further elaboration. Are these features complementary, enhancing the overall detection accuracy, or redundant?

Response: We deeply appreciate your precious comment regarding the relationship between the multidimensional features. Upon reflection, we recognize that our initial explanation in the manuscript was not sufficiently clear, which may have led to ambiguity about whether these features are complementary or redundant. To address this, we have revised the manuscript to provide a more detailed discussion of the complementary relationship between multidimensional features and their role in improving detection accuracy. Specifically, we emphasize that multidimensional features complement each other, and through their complementary design, MDFM can comprehensively characterize border binaries. These changes are on Pages 10-11, Lines 374–389.

Revision text:

The 15-dimensional features characterize border binaries from different perspectives and exhibit the following complementary relationships:

Program-level features (V0 and V1) provide a global perspective, aiding in identifying the binary’s overall behavior and semantics. For example, V0 can preliminarily assess whether the binary may be a border binary by counting the number of APIs related to external input in the function import table.

Function-level features (V2~V14) provide a localized perspective, aiding in identifying specific network parsing functions.

Feature weight allocation. In the feature of border binaries, function-level features are assigned higher weights, reflecting the importance of network parsing functions in border binaries, while program-level features act as supplementary components to ensure that the model encompasses all possible border binaries.

By utilizing the complementary design of multidimensional features, MDFM accurately and comprehensively characterizes border binary features, enhancing the accuracy and robustness of border binary detection. A detailed list of the feature vectors and their corresponding descriptions is provided in Table 2.

Reviewer #2

Reviewer Comments on BBDetector: Intelligent Border Binary Detection in IoT Device Firmware Based on a Multi-Dimensional Feature Model

Comment 1: The abstract should provide a concise summary of the research problem, methods, results, and significance. Currently, it lacks clarity on the specific contributions of the study.

Response: It’s great that you’ve considered the feedback and added the contributions to the abstract. Clearly outlining the specific contributions of the study is crucial for readers to grasp the research value quickly. We have added the contribution at the end of the abstract as your suggestion.

Revision text:

BBDetector provides an accurate method for border binary detection in IoT firmware security analysis, significantly enhancing the pertinence of vulnerability detection, reducing the complexity of firmware security analysis, and providing essential technical support for improving IoT device security.

Comment 2: Expand the literature review to include more recent studies on Border Binary detection in IoT devices. Highlight how this proposed research work differentiates from existing research.

Response: We sincerely appreciate your valuable comment regarding the literature review. We have expanded our literature review section to include more recent studies on border binary detection in IoT devices and to highlight how our proposed research differentiates from existing research. We believe this will further strengthen the credibility and rigor of our study.

Revision text:

Given the extensive length of the revisions, it is challenging to present the complete modifications in the response letter. The specific revised sections can be found on Page 4, Lines 116-122 and Pages 4-5, Lines 160-191.

Comment 3: Provide a detailed explanation of the ensemble learning, combining extreme gradient boosting, light gradient boosting machine, and categorical boosting as base learners with random forest as the meta-learner. Include mathematical formulations and pseudocode for better understanding.

Response: Thanks for the careful review of the ensemble learning framework in our study. We sincerely apologize for the oversight in not initially providing a detailed explanation of the ensemble learning method, which may have posed challenges for readers in fully understanding the framework. In response to your comments, we have revised the manuscript by adding mathematical formulations for the ensemble learning framework and providing a detailed explanation of the training and prediction phases through the pseudocode in Algorithm 2. These additions aim to enhance the clarity and comprehensibility of our approach, ensuring that readers can better grasp the integration of extreme gradient boosting, light gradient boosting machine, and categorical boosting as base learners, with random forest as the meta-learner.

Revision text:

To maintain clarity and conciseness in the response letter, we have not included the complete modifications here due to their extensive length. The specific revised sections are in Pages 14-16, Lines 515-558.

Comment 4: Authors should include the hyperparameters of the Gradient boosting model, light gradient boosting machine.

Response: We sincerely thank the reviewer for their constructive comments. In the revised manuscript, we have added the parameter settings for XLC-R, including the hyperparameters of the extreme gradient boosting model (XGBoost) and light gradient boosting machine (LightGBM). To obtain the optimal detection model, we employed a grid search technique to determine these hyperparameters. Detailed information about the hyperparameters can be found in Section 6.2 (Experimental setup) on Page 17, Lines 581-585 of the revised manuscript.

Revision text:

The configuration adopted by XLC-R was as follows: XGBoost used 100 estimators, with an evaluation metric of logloss, a learning rate of 0.1, and a random state of 42; LightGBM used 100 estimators, with a learning rate of 0.1, a maximum depth of -1 and a random state of 42; CatBoost used 100 estimators, with a random state of 42; and RF used 20 estimators, with a maximum depth of 4, and a random state of 42.

Comment 5: Authors should include the hyperparameters of light gradient boosting machine.

Response: Thank you for your advice. We have taken them seriously and added relevant content about the Light Gradient Boosting Machine (LightGBM) in the revised manuscript, particularly providing a detailed explanation of its hyperparameters, including the specific settings for key hyperparameters such as the learning rate and maximum depth. You can find these details in Section 6.2 (Experimental setup). We believe that these additions have made the study more comprehensive and rigorous.

Revision text:

The specific details regarding this point are provided in the revision text under Comment 4. To avoid redundancy, we have not repeated them here.

Comm

---

## [Decision Letter · Decision Letter 1]

26 Jun 2025

PONE-D-24-50757R1BBDetector: Intelligent border binary detection in IoT device firmware based on a multidimensional feature model

PLOS ONE

Dear Dr. Zhang,

Thank you for submitting your manuscript to PLOS ONE. After careful consideration, we feel that it has merit but does not fully meet PLOS ONE’s publication criteria as it currently stands. Therefore, we invite you to submit a revised version of the manuscript that addresses the points raised during the review process.

Academic Editor:Dear author, You are requested to revise you manuscript as per the comments. Insufficient Contextualization: The paper would benefit from a stronger connection to the existing literature. The introduction and related work section should explicitly discuss how BBDetector builds upon, differs from, or improves upon recent advances in IoT firmware security and border binary detection.Limited Recent References: The current reference list appears to lack sufficient representation of recent publications (2023-2025) in the specific area of IoT firmware analysis and vulnerability detection. This limits the paper's ability to demonstrate its novelty and relevance.Suggested Citations: The authors should consider incorporating the following recent articles and others identified through a more comprehensive literature search. 1. Xia, J., Li, S., Huang, J., Yang, Z., Jaimoukha, I. M.,... Gündüz, D. (2023). Metalearning-Based Alternating Minimization Algorithm for Nonconvex Optimization. IEEE Transactions on Neural Networks and Learning Systems, 34(9), 5366-5380. doi: 10.1109/TNNLS.2022.31656272. Zhou, W., Xia, C., Wang, T., Liang, X., Lin, W., Li, X.,... Zhang, S. (2025). HIDIM: A novel framework of network intrusion detection for hierarchical dependency and class imbalance. Computers & Security, 148, 104155. doi: https://doi.org/10.1016/j.cose.2024.1041553. Zhou, Z., Li, Z., Zhou, W., Chi, N., Zhang, J.,... Dai, Q. (2025). Resource-Saving and High-Robustness Image Sensing Based on Binary Optical Computing. Laser & Photonics Reviews, 19(7), 2400936. doi: https://doi.org/10.1002/lpor.2024009364. Huang, S., Sun, C., & Pompili, D. (2025). Meta-ETI: Meta-Reinforcement Learning with Explicit Task Inference for UAV-IoT Coverage. IEEE Internet of Things Journal. doi: 10.1109/JIOT.2025.35538085. Zhang, Z., Liu, Z., Martin, A., & Zhou, K. (2023). BSC: Belief Shift Clustering. IEEE Transactions on Systems, Man, and Cybernetics: Systems, 53(3), 1748-1760. doi: 10.1109/TSMC.2022.32053656. Liu, Y., Huo, M., Li, M., He, L., & Qi, N. (2025). Establishing a Digital Twin Diagnostic Model Based on Cross-Device Transfer Learning. IEEE Transactions on Instrumentation and Measurement, 74, 1-10. doi: 10.1109/TIM.2025.35629737. Zhang, H., Xu, Y., Luo, R., & Mao, Y. (2023). Fast GNSS acquisition algorithm based on SFFT with high noise immunity. China Communications, 20(5), 70-83. doi: 10.23919/JCC.2023.00.0068. Qiao, Y., Lü, J., Wang, T., Liu, K., Zhang, B.,... Snoussi, H. (2024). A Multihead Attention Self-Supervised Representation Model for Industrial Sensors Anomaly Detection. IEEE Transactions on Industrial Informatics, 20(2), 2190-2199. doi: 10.1109/TII.2023.32803379. Zhang, M., Wei, E., Berry, R., & Huang, J. (2024). Age-Dependent Differential Privacy. IEEE Transactions on Information Theory, 70(2), 1300-1319. doi: 10.1109/TIT.2023.334014710. Li, X., Lu, Z., Yuan, M., Liu, W., Wang, F., Yu, Y.,... Liu, P. (2024). Tradeoff of Code Estimation Error Rate and Terminal Gain in SCER Attack. IEEE Transactions on Instrumentation and Measurement, 73, 1-12. doi: 10.1109/TIM.2024.340680711. Jiang, N., Feng, Q., Yang, X., He, J., & Li, B. (2025). The octonion linear canonical transform: Properties and applications. Chaos, Solitons & Fractals, 192, 116039. doi: https://doi.org/10.1016/j.chaos.2025.116039Please ensure that your decision is justified on PLOS ONE’s publication criteria and not, for example, on novelty or perceived impact.

We look forward to receiving your revised manuscript.

Kind regards,

Mudassir Khan, Ph.D

Academic Editor

PLOS ONE

Journal Requirements:

Additional Editor Comments (if provided):

Dear author,

You are requested to revise you manuscript as per the comments.

Insufficient Contextualization: The paper would benefit from a stronger connection to the existing literature. The introduction and related work section should explicitly discuss how BBDetector builds upon, differs from, or improves upon recent advances in IoT firmware security and border binary detection.

Limited Recent References: The current reference list appears to lack sufficient representation of recent publications (2023-2025) in the specific area of IoT firmware analysis and vulnerability detection. This limits the paper's ability to demonstrate its novelty and relevance.

Suggested Citations: The authors should consider incorporating the following recent articles and others identified through a more comprehensive literature search.

1. Xia, J., Li, S., Huang, J., Yang, Z., Jaimoukha, I. M.,... Gündüz, D. (2023). Metalearning-Based Alternating Minimization Algorithm for Nonconvex Optimization. IEEE Transactions on Neural Networks and Learning Systems, 34(9), 5366-5380. doi: 10.1109/TNNLS.2022.3165627

2. Zhou, W., Xia, C., Wang, T., Liang, X., Lin, W., Li, X.,... Zhang, S. (2025). HIDIM: A novel framework of network intrusion detection for hierarchical dependency and class imbalance. Computers & Security, 148, 104155. doi: https://doi.org/10.1016/j.cose.2024.104155

3. Zhou, Z., Li, Z., Zhou, W., Chi, N., Zhang, J.,... Dai, Q. (2025). Resource-Saving and High-Robustness Image Sensing Based on Binary Optical Computing. Laser & Photonics Reviews, 19(7), 2400936. doi: https://doi.org/10.1002/lpor.202400936

4. Huang, S., Sun, C., & Pompili, D. (2025). Meta-ETI: Meta-Reinforcement Learning with Explicit Task Inference for UAV-IoT Coverage. IEEE Internet of Things Journal. doi: 10.1109/JIOT.2025.3553808

5. Zhang, Z., Liu, Z., Martin, A., & Zhou, K. (2023). BSC: Belief Shift Clustering. IEEE Transactions on Systems, Man, and Cybernetics: Systems, 53(3), 1748-1760. doi: 10.1109/TSMC.2022.3205365

6. Liu, Y., Huo, M., Li, M., He, L., & Qi, N. (2025). Establishing a Digital Twin Diagnostic Model Based on Cross-Device Transfer Learning. IEEE Transactions on Instrumentation and Measurement, 74, 1-10. doi: 10.1109/TIM.2025.3562973

7. Zhang, H., Xu, Y., Luo, R., & Mao, Y. (2023). Fast GNSS acquisition algorithm based on SFFT with high noise immunity. China Communications, 20(5), 70-83. doi: 10.23919/JCC.2023.00.006

8. Qiao, Y., Lü, J., Wang, T., Liu, K., Zhang, B.,... Snoussi, H. (2024). A Multihead Attention Self-Supervised Representation Model for Industrial Sensors Anomaly Detection. IEEE Transactions on Industrial Informatics, 20(2), 2190-2199. doi: 10.1109/TII.2023.3280337

9. Zhang, M., Wei, E., Berry, R., & Huang, J. (2024). Age-Dependent Differential Privacy. IEEE Transactions on Information Theory, 70(2), 1300-1319. doi: 10.1109/TIT.2023.3340147

10. Li, X., Lu, Z., Yuan, M., Liu, W., Wang, F., Yu, Y.,... Liu, P. (2024). Tradeoff of Code Estimation Error Rate and Terminal Gain in SCER Attack. IEEE Transactions on Instrumentation and Measurement, 73, 1-12. doi: 10.1109/TIM.2024.3406807

11. Jiang, N., Feng, Q., Yang, X., He, J., & Li, B. (2025). The octonion linear canonical transform: Properties and applications. Chaos, Solitons & Fractals, 192, 116039. doi: https://doi.org/10.1016/j.chaos.2025.116039

Reviewers' comments:

Reviewer's Responses to Questions

**Comments to the Author**

1. If the authors have adequately addressed your comments raised in a previous round of review and you feel that this manuscript is now acceptable for publication, you may indicate that here to bypass the “Comments to the Author” section, enter your conflict of interest statement in the “Confidential to Editor” section, and submit your "Accept" recommendation.

Reviewer #1: All comments have been addressed

Reviewer #2: All comments have been addressed

Reviewer #3: All comments have been addressed

2. Is the manuscript technically sound, and do the data support the conclusions?

Reviewer #1: Yes

Reviewer #2: Yes

Reviewer #3: Yes

3. Has the statistical analysis been performed appropriately and rigorously? 

Reviewer #1: Yes

Reviewer #2: Yes

Reviewer #3: N/A

4. Have the authors made all data underlying the findings in their manuscript fully available?

Reviewer #1: Yes

Reviewer #2: Yes

Reviewer #3: Yes

5. Is the manuscript presented in an intelligible fashion and written in standard English?

Reviewer #1: Yes

Reviewer #2: Yes

Reviewer #3: Yes

6. Review Comments to the Author

Reviewer #1: The authors have addressed all my comments, and this paper is improved significantly, so I think this paper can be accepted for publication in this journal.

Reviewer #2: The authors have addressed all the comments. The authors have addressed all the comments. The authors have addressed all the comments. The authors have addressed all the comments. The authors have addressed all the comments. The authors have addressed all the comments. The authors have addressed all the comments. The authors have addressed all the comments. The authors have addressed all the comments.

Reviewer #3: Authors did well to address comments. They addressed all comments highlighted previously and gave proper response to each alongwith changes made.

7. PLOS authors have the option to publish the peer review history of their article (what does this mean?). If published, this will include your full peer review and any attached files.

Reviewer #1: No

Reviewer #2: **Yes: **Yakub Kayode Saheed

Reviewer #3: No

---

## [Author Response · Author response to Decision Letter 2]

9 Jul 2025

We prepared docx file for answering each reviewer's comments. In brief. we revised the following contents.

1. Modified the introduction and related work sections to more clearly explain how BBDetector differs from the latest methods in this field.

2. Updated the reference list by adding more recent related studies.

---

## [Editor Report · Decision Letter 2]

17 Jul 2025

BBDetector: Intelligent border binary detection in IoT device firmware based on a multidimensional feature model

PONE-D-24-50757R2

Dear Author,

We’re pleased to inform you that your manuscript has been judged scientifically suitable for publication and will be formally accepted for publication once it meets all outstanding technical requirements.

Kind regards,

Mudassir Khan, Ph.D

Academic Editor

PLOS ONE

Additional Editor Comments (optional):

Thanks to the authors for the detailed response and additions. I read through the comments and skimmed the revised PDF, and the updates significantly improved the paper. I would be happy to recommend this paper for publication.
---

## [Editor Report · Acceptance letter]

PONE-D-24-50757R2

PLOS ONE

Dear Dr. Zhang,

I'm pleased to inform you that your manuscript has been deemed suitable for publication in PLOS ONE. Congratulations! Your manuscript is now being handed over to our production team.

Kind regards,

on behalf of

Dr. Mudassir Khan

Academic Editor

PLOS ONE